# Contrasting influences of biogeophysical and biogeochemical impacts of historical land use on global economic inequality

Shu Liu[1], Yong Wang [1✉], Guang J. Zhang [2], Linyi Wei[1], Bin Wang[1,3,4] & Le Yu [1]

Climate change has significant implications for macro-economic growth. The impacts of greenhouse gases and anthropogenic aerosols on economies via altered annual mean temperature (AMT) have been studied. However, the economic impact of land-use and land-cover change (LULCC) is still unknown because it has both biogeochemical and biogeophysical impacts on temperature and the latter differs in latitudes and disturbed land surface types. In this work, based on multi-model simulations from the Coupled Model Intercomparison Project Phase 6, contrasting influences of biogeochemical and biogeophysical impacts of historical (1850–2014) LULCC on economies are found. Their combined effects on AMT result in warming in most countries, which harms developing economies in warm climates but benefits developed economies in cold climates. Thus, global economic inequality is increased. Besides the increased AMT by the combined effects, day-to-day temperature variability is enhanced in developing economies but reduced in developed economies, which further deteriorates global economic inequality.

[1] Department of Earth System Science, Ministry of Education Key Laboratory for Earth System Modeling, Institute for Global Change Studies, Tsinghua University, Beijing 100084, China. [2] Scripps Institution of Oceanography, La Jolla, CA, USA. [3] State Key Laboratory of Numerical Modeling for Atmospheric Sciences and Geophysical Fluid Dynamics, Institute of Atmospheric Physics, Chinese Academy of Sciences, Beijing, China. [4] College of Earth and Planetary Sciences, University of Chinese Academy of Sciences, Beijing, China. ✉email: yongw@mail.tsinghua.edu.cn

Owing to humans' growing demands for food, fiber, and shelter, the Earth's land surface has been dramatically disturbed by human activities, with extensive natural landscapes being converted to human-dominated lands, such as cropland, grazing land, and urban impervious surfaces[1–4]. At the same time, land-use management (e.g., irrigation, fertilization, and wood harvest) also perturbs land surfaces, with 42–58% of them managed[5]. In the coming decades, the demands for food and energy will likely surge because of predicted increases in the global population and productivity. Therefore, anthropogenic land-use and land-cover change (LULCC) is expected to intensify further to meet future increasing demands[6].

Land-use practices providing critical natural resources are essential for human welfare, but some excessive forms (e.g., overgrazing and overcutting) are degrading the ecosystem (e.g., soil erosion and land desertification), which undermine ecosystem services, decrease economic and social benefits, and threaten the long-term sustainability of human societies[7,8]. Historical LULCC has also been recognized as a key driver of anthropogenic climate change, according to the Fifth Assessment Report of the Intergovernmental Panel on Climate Change (IPCC)[9,10] because it regulates the exchanges of momentum, energy, water, and carbon with the atmosphere through biogeophysical (BGP) and biogeochemical (BGC) processes[11–16]. Global net BGC impacts of LULCC have contributed to approximately 25% of the historical increase in $CO_2$ emissions since the Industrial Revolution[10]. Owing to the spatial heterogeneity of the land surface, the BGP impact of LULCC on temperature spatially is not as homogeneous as warming induced by greenhouse gases and cooling resulting from aerosols[9]. It varies in latitudes and disturbed land surface types[17,18]. For instance, the BGP impact of deforestation leads to warming in the tropics by reducing evapotranspiration but cooling in the extratropics because of increased surface albedo[17–19]. In general, the BGP impact of global LULCC contributes to global radiative forcing of −0.15 (−0.25 to −0.05) W/m² relative to the pre-industrial level, masking some of the warming induced by greenhouse gases[9]. When BGP and BGC processes are considered together, up to 40% (±16%) of the present-day global anthropogenic warming can be attributed to historical LULCC[20].

Recent research has shown that climate change can affect economic development by influencing agricultural yields, energy supply, labor productivity, and human health[21]. Economic impacts of climate change were predominantly studied by integrated assessment models, which parameterize a series of physical and socioeconomic relationships of climate variables and economic indicators[22]. A growing number of studies have used historical climate observations and socioeconomic data to empirically estimate the impact of climate change on economic growth[23–28]. Observational evidence suggests significant nonlinear responses of economic growth to changes in annual mean temperature across countries, with the per capita gross domestic product (GDP) growth rate peaking at optimal temperature and declining at higher or lower temperatures[24,27]. This nonlinear response is referred to as the "temperature–growth response function". According to the empirical nonlinear temperature–growth response function, it has been noted that global historical and projected anthropogenic climate warming damage developing economies over the tropics but benefit developed economies in cooler climates, thus increasing global economic inequality[27,28]. In contrast, the cooling caused by historical anthropogenic aerosol emissions, which offsets approximately one-third of the warming from the increases in greenhouse gases, probably reduces global economic inequality[29].

Previous studies using the temperature–growth response function only account for the economic impact of the change in annual mean temperature. Recent work revealed that day-to-day temperature variability has a greater economic influence than the annual mean temperature because dramatic economic losses occur on extremely hot and cold days[25,30,31]. An extra degree Celsius of temperature variability reduces the growth rate of GDP per capita by 5% on average, featuring higher vulnerability in low-latitude and low-income countries[25]. Historical LULCC has been found to contribute significantly to the observed greater day-to-day temperature variability as well[13,16,32,33], in addition to its impact on annual mean temperature[9,12,18].

It has been documented that long-term economic sustainable development is limited due to substantial disruption to the ecosystem from historical LULCC despite short-term regional economic growth with the gains of agricultural/industrial commodities[34–37]. Nonetheless, the economic impact of historical LULCC through climate feedback is still unclear. This is because the climatic impacts of LULCC on economies are much more complicated than those of greenhouse gases and anthropogenic aerosols. LULCC has both BGP and BGC impacts on climate, and the BGP impacts vary in latitudes and land surface types. Furthermore, besides the economic effects of the annual-mean temperature change induced by historical LULCC, the additionally imposed effects on economies from day-to-day temperature variability changes need to be figured out as well.

To fill this gap, we investigate country-level cumulative economic impacts of changes in annual mean surface air temperature (SAT) and day-to-day SAT variability from 1993 to 2012 due to the BGP and BGC impacts of historical LULCC since 1850. The economic impact is measured by GDP per capita, showing the value of goods and services produced within a country in a year per person. Details of our analytic approach are documented in the Methods. In summary, the country-level SAT and economic conditions of the two worlds are compared: one is the factual world with LULCC represented by observations[38–41], and the other is the counterfactual world, which is identical to the former except without the BGP and BGC impacts of LULCC. The BGP and BGC impacts on mean SAT and day-to-day SAT variability are estimated by multi-model climate simulation experiments from the Coupled Model Intercomparison Project Phase 6 (CMIP6) (Supplementary Table 1) (see Methods for details)[20,42–45]. The economic impact of annual mean SAT changes is estimated following the temperature–growth response function with 1,000 members generated by bootstrapping[27]. The economic impact of day-to-day SAT variability changes is quantified according to a damage function describing the country-level responses of economic growth to every extra degree Celsius of day-to-day temperature variability[25]. Here, we show the contrasting influences of BGP and BGC impacts of historical LULCC on economies. Their combined effects on annual mean SAT and day-to-day SAT variability harm economically disadvantaged countries but benefit economically advanced countries, thus increasing global economic inequality.

## Results

**Historical LULCC since the industrial revolution.** Figure 1 shows the global historical LULCC from 1850 to 2014 based on the Land-Use Harmonization 2 (LUH2) dataset, which is the land-use forcing for the CMIP6 "historical" experiment (see Methods)[4,43,46]. The global fractional coverage of primary vegetation (including forest and non-forest) in 2014 was reduced by 89.2% relative to that in 1850, while secondary vegetation, cropland, grazing land, and urban land exhibited increases of 81.2%, 172.6%, 176.9%, and 1349.5%, respectively (Fig. 1a). Humans have disturbed a majority of primary vegetation globally (Fig. 2). Secondary vegetation showed significant increases over mid-latitude Eurasia and the eastern USA. Cropland expansion was mainly distributed around the Great Lakes of North America,

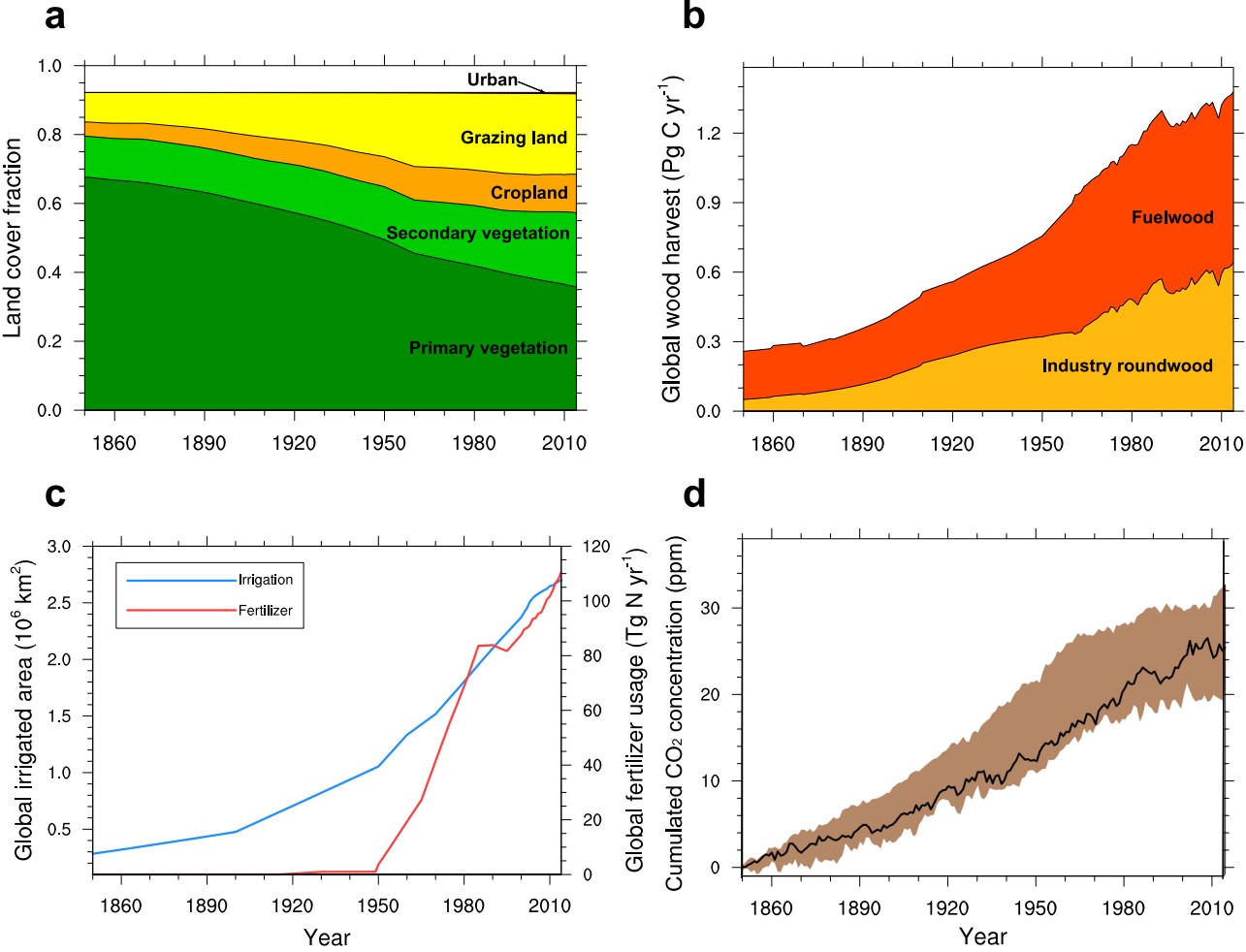

**Fig. 1 Global historical land-use and land-cover change (LULCC) and the resulting cumulative increase in atmospheric CO₂ concentrations from 1850 to 2014. a** Land-cover fractions (relative to global land except for Antarctica) of primary vegetation (dark green), secondary vegetation (light green), cropland (orange), grazing land (yellow), and urban land (pink). The fractions of the five land-cover types add up to less than 1 due to the coverage of ice and water (shown in the white area). **b** Annual global wood harvest (Pg C yr⁻¹) for fuelwood (red) and industrial roundwood (yellow). **c** Global irrigated area (blue line, 10⁶ km²) and annual fertilizer usage (red line, Tg N yr⁻¹). **a–c** are based on the LUH2 dataset[4,43,46]. **d** LULCC-induced cumulative increase in atmospheric CO₂ concentrations (ppm). The black line and corresponding brown shading denote the median and 25–75th percentile range of ensemble members.

south of the Amazon, central Africa, north of the Caspian Sea, and India. The expansion of grazing land was more significant, mainly located in the USA, south of the Amazon, central and southern Africa, central Asia, and Australia. Although the area of urban land is smaller than the other four types of land cover, its relative change was the largest, mainly over the eastern USA, Europe, and China.

For human land-use management, owing to the increased global demand for timber, the annual global wood harvest (disaggregated into fuelwood and industrial-use roundwood) was growing (Fig. 1b), mainly from the Amazon, middle Africa, and South Asia (Supplementary Fig. 1a). Along with cropland expansion, the global irrigated cropland area increased from 0.28 to 2.70 million km² during 1850–2014 (Fig. 1c). The expansion of this mostly occurred in China, India, Southeast Asia, and the Middle East (Supplementary Fig. 1b). The global annual usage of synthetic nitrogen fertilizer on croplands was zero before 1915 and began to increase rapidly after 1950 (Fig. 1c). By 2014, it had risen to 110.6 Tg N per year, mainly in China, India, the USA, and Europe (Supplementary Fig. 1c). LULCC, on the one hand, influences surface energy and water balance by altering surface properties (e.g., albedo and Bowen ratio) via the BGP

processes[9]. On the other hand, it leads to additional CO₂ emissions via the BGC processes[10]. By 2014, the LULCC contribution to the atmospheric CO₂ concentration is approximately 25.3 ppm (Fig. 1d), an increase of 9% relative to the atmospheric CO₂ concentration in 1850, in line with previous studies[20,45].

**Annual mean SAT and day-to-day SAT variability changes.** We focus on the impacts of historical LULCC on SAT and associated economies during 1993–2012 given that the socioeconomic data of most countries are available for this period. Consistent with previous studies[17–19], the BGP impact of deforestation leads to significant cooling in North America and Eurasia mainly because of increased surface albedo but warming over the Amazon and central Africa driven by reduced evapotranspiration (Fig. 3a). The warming over Greenland where minor LULCC exists is related to the atmospheric circulation changes induced by non-local LULCC. Strong cooling at midlatitudes in the Northern Hemisphere weakens the upper-troposphere westerly jet and thus results in warm advection in Greenland[47]. The BGC impact of LULCC leads to warming globally due to the emission of CO₂, a long-lived greenhouse gas well mixed in the atmosphere (Fig. 3b). Over most

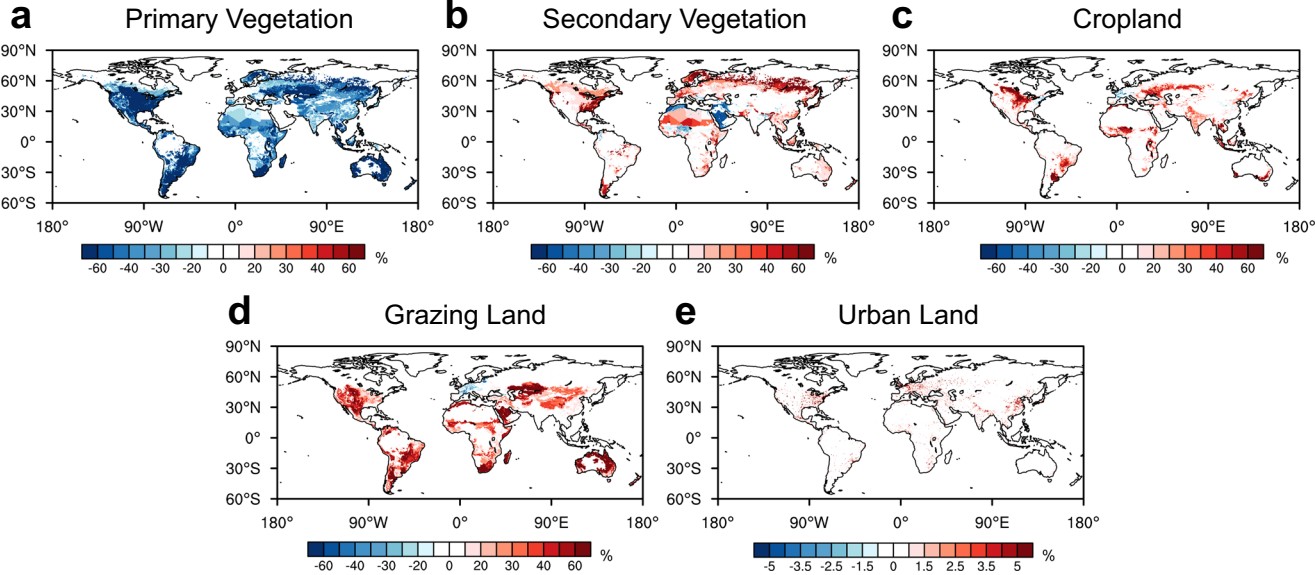

**Fig. 2 Spatial patterns of historical land-use and land-cover change (LULCC) from 1850 to 2014 (%, 2014 minus 1850). a** Primary vegetation. **b** Secondary vegetation. **c** Cropland. **d** Grazing land. **e** Urban land. The historical LULCC is based on the LUH2 dataset[4,43,46].

continents of the world, BGC-induced warming dominates the combined SAT changes, except the central and eastern parts of North America, Central Asia, and East Europe (Fig. 3c)[20,44,45].

Day-to-day SAT variability reflects the magnitude of daily temperature fluctuations. Greater SAT variability implies stronger daily heat or/and cold extremes[48]. As shown in Fig. 3d–f, the BGC impact plays a leading role in LULCC-induced SAT variability changes. The SAT variability is decreased at high latitudes in the Northern Hemisphere (north of 60 °N) but increased in the tropics and subtropics (30 °S–30 °N) (Fig. 3f). It has been documented that the warming due to increased atmospheric $CO_2$ concentrations leads to Arctic sea-ice loss and thus reduces the meridional temperature gradient in the extratropics due to Arctic amplification. This further weakens synoptic wave activities and results in decreased day-to-day SAT variability there[49,50]. The increased SAT variability in the tropics is mainly caused by soil drying in response to the warming from the increased $CO_2$ concentrations[49–51].

**Economic impacts of annual mean SAT changes.** We first estimate the economic impacts of LULCC-induced annual mean SAT changes, according to the bootstrapped temperature–growth response function[27]. Figure 4 shows the combined impacts of LULCC on economies. For warm countries with annual mean SAT warmer than the temperature optimum (mostly between 30 °S and 30 °N), LULCC-induced warming makes country-level temperatures deviate further from the temperature optimum, thus damaging the economies there (e.g., Zambia, India, and Saudi Arabia). In contrast, the warming benefits the economic growth for cool countries whose annual mean SAT is colder than the temperature optimum (mostly north of 45 °N) because it makes country-level temperatures closer to the temperature optimum (e.g., Iceland and the UK). For mid-latitude countries with annual mean SAT close to the temperature optimum (e.g., the USA and China), economic growth is insensitive to annual mean SAT changes there. These changes are regulated by the BGC impact of LULCC due to its dominant role in annual mean temperature changes (Supplementary Fig. 2a), except Canada, which is controlled by the BGP-induced cooling (Supplementary Fig. 3a).

The opposite economic impacts of LULCC-induced warming on warm and cool countries may aggravate global economic inequality (Fig. 4) because most economically disadvantaged countries are in the low latitudes of the warm climate, but economically advanced countries generally situate in temperate and cool climates[27–29] (Supplementary Fig. 4). For example, for India (Fig. 4c, d), a developing country with a warm climate, the net warming impact of LULCC decreases the growth rate of GDP per capita annually, leading to cumulative economic damage of up to −6.35% (−0.69 to −12.20% in the 25–75th range) in 2012. In comparison, the UK, a developed country with a cool climate, shows annual increases in the growth rate of GDP per capita because of warming, resulting in cumulative economic benefits of up to +1.60% (+0.34 to +3.91% in the 25–75th range) in 2012 (Fig. 4e, f). At the global scale, LULCC-induced warming decreases annual economic growth, leading to cumulative economic damage of approximately −1.30% (−3.02 to −0.11% in the 25th–75th range) in 2012 (Fig. 4g, h). Note that although the BGC effect dominates the overall warming (Supplementary Fig. 2g, h), the BGP-induced cooling favors global annual economic growth, with global cumulative economic gains of approximately +0.88% (+0.10 to +2.19% in the 25th–75th range) in 2012 (Supplementary Fig. 3g, h; Supplementary Table 2).

Figure 5 shows the global country-level cumulative economic impacts of the annual mean SAT changes from 1993 to 2012 caused by the respective BGP and BGC impacts and their combination of historical LULCC. Contrasting economic influences of individual BGP and BGC impacts of LULCC are found owing to their opposite influences on annual mean SAT (Fig. 3). Most low-latitude countries with warm climates (e.g., the majority of African and Southeast Asian countries) are experiencing positive economic impacts from the BGP cooling, but negative economic impacts from the BGC warming (Fig. 5a, c). Many cool climate countries in the high latitudes of the Northern Hemisphere (e.g., Russia, Canada, and Norway) experience damage from the BGP cooling but benefit from the BGC warming. The combined impact of LULCC on economies is generally controlled by the BGC warming (Fig. 5e), except for some countries with cool climates over northern midlatitudes cooled by the BGP effect (Fig. 3c), which decreases their economic growth (e.g., Canada, Sweden, and Finland).

The individual BGP and BGC impacts and their combination on economies in countries of different economic conditions are

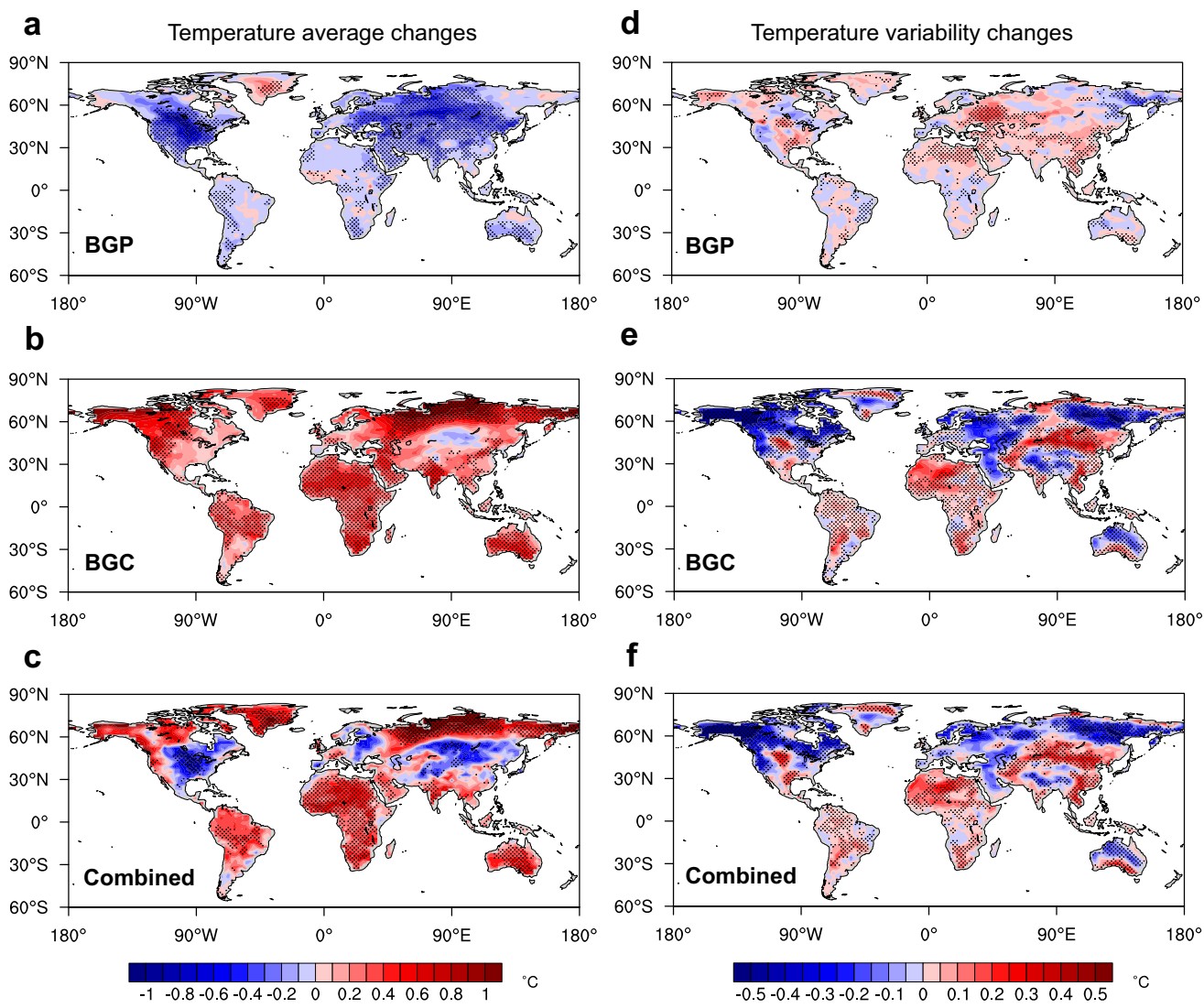

**Fig. 3 Spatial patterns of changes in annual mean surface air temperature (SAT) and day-to-day SAT variability during 1993–2012 due to biogeophysical (BGP) and biogeochemical (BGC) impacts of historical land-use and land-cover change. a–c** The ensemble median of annual mean SAT changes (°C) induced by (**a**) BGP, (**b**) BGC, and (**c**) their combined impacts. **d–f** The ensemble median of annual mean day-to-day SAT variability changes (°C) induced by (**d**) BGP, (**e**) BGC, and (**f**) their combined impacts. Dots indicate where more than two-thirds of members agree on the sign of response.

illustrated in Fig. 6. The combined impact of LULCC largely modulated by the BGC warming benefits many economically advanced countries with higher GDP per capita (bars shaded in the redder color on the right), but damages many economically disadvantaged countries with lower GDP per capita (bars shaded in the bluer color on the left). In contrast, the BGP cooling impact of LULCC, albeit canceled out by the BGC-dominated warming, harms many economically advanced countries (e.g., Russia, Canada, and Norway) mostly in cool climates but favors many economically disadvantaged countries (e.g., Indonesia, Egypt, and India) in the warm tropics.

We further use common measures of economic inequality (the 80:20 and 90:10 ratios of the population-weighted percentile of GDP per capita[52]) to quantitatively estimate the impacts of LULCC on global economic inequality (Fig. 7). Both 80:20 and 90:10 ratios reflect the global economic gap between the top economically advanced country and the bottom economically disadvantaged country (see Methods). Compared with the counterfactual world without the combined impacts of LULCC, LULCC increases the economic gap almost annually from 1993 to 2012 (Fig. 7c). By 2012, the 80:20 and 90:10 ratios of the global economic gap are

significantly increased by +5.10% (+1.18% to +12.75% in the 25–75th range) and +2.64% (−0.80% to +5.23% in the 25–75th range), respectively (Fig. 7f). The increased economic inequality is due to the BGC warming impact (Fig. 7b, e), which is partly offset by the decreased economic inequality induced by the BGP cooling impact (Fig. 7a, d). In summary, although the overall global economic inequality among countries has decreased over the past decades, our results show that the climate impacts of historical LULCC on annual mean temperature negatively impact the decrease.

**Economic impacts of day-to-day SAT variability changes**. The annual mean SAT changes capture a fraction of SAT impacts on economies. In addition to the annual mean SAT changes, the changes in day-to-day SAT variability have a greater economic influence[25]. With the global maps of annual mean day-to-day SAT variability changes induced by BGC and BGP impacts of historical LULCC and their combination (Fig. 3), we investigate their corresponding influences on economies. Overall, the signs of the BGC and BGP impacts on economies are opposite in many

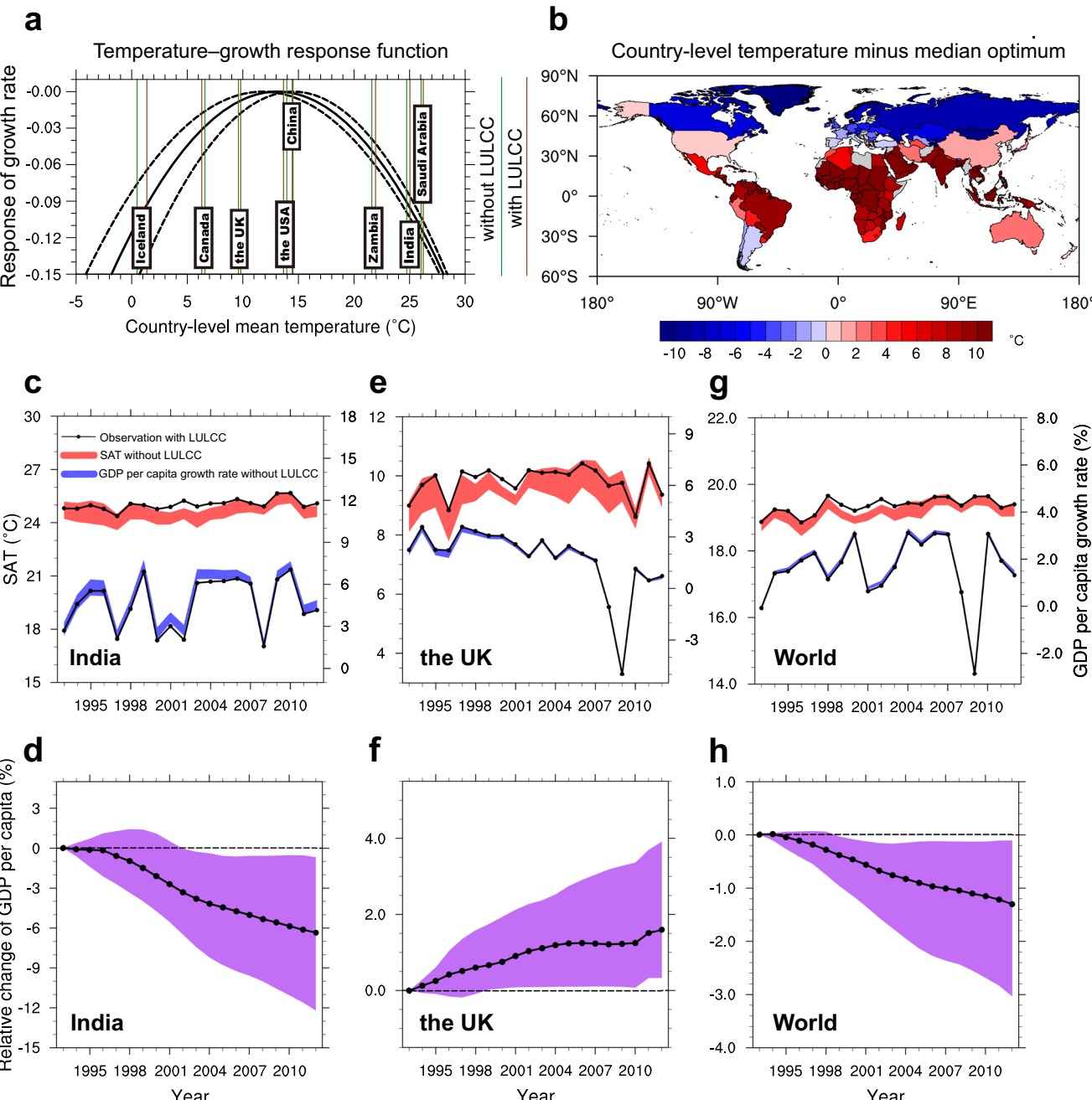

**Fig. 4 Country-level annual-mean surface air temperature (SAT) changes and associated economic impacts during 1993–2012 due to the combined impacts of historical land-use and land-cover change (LULCC). a** Bootstrapped temperature–growth response function represented by 25th (left dashed curve), 50th (solid curve), and 75th (right dashed curve) percentiles of 1000 members of the temperature optimum[27]. Vertical lines overlaid on the curves are annual mean temperatures from factual observations with LULCC (brown) and the counterfactual world without LULCC (green) for some representative countries. **b** Spatial pattern of the difference between the country-level annual mean temperature and median temperature optimum. Countries and regions with missing values are shaded in gray. **c, e, g** The 25–75th percentile range of SAT (red shading, in °C) and GDP per capita growth rate (blue shading, in %) in the counterfactual world without LULCC, both with corresponding observations in the factual world with LULCC (black dotted line) for (**c**) India, (**e**) the UK, and (**g**) the world. **d, f, h** Relative changes in GDP per capita (%) induced by LULCC for (**d**) India, (**f**) the UK, and (**h**) the world. The black dotted line and corresponding purple shading indicate the median and 25–75th percentile range of ensemble members.

countries (Fig. 8a, c). The combined impact on economies is regulated by the BGC impact due to its dominant role in affecting the day-to-day SAT changes (Figs. 3f and 8e). In comparison with countries in midlatitudes and high latitudes, low-latitude countries with smaller seasonal temperature variations are damaged more given the increases in day-to-day SAT variability because they have more difficulty in adapting to greater temperature fluctuations[25] (Fig. 8e). In addition, low-latitude countries with

developing economies (Supplementary Fig. 4) are more vulnerable to the negative economic impact of the increased day-to-day SAT variability because of less insurance and risk-management practices[25]. Therefore, many low-latitude and low-income countries (e.g., the majority of African and Southeast Asian countries) are greatly damaged from the increased day-to-day SAT variability imposed by the combined impact of LULCC (Fig. 8e). In contrast, the economies of many rich countries in the

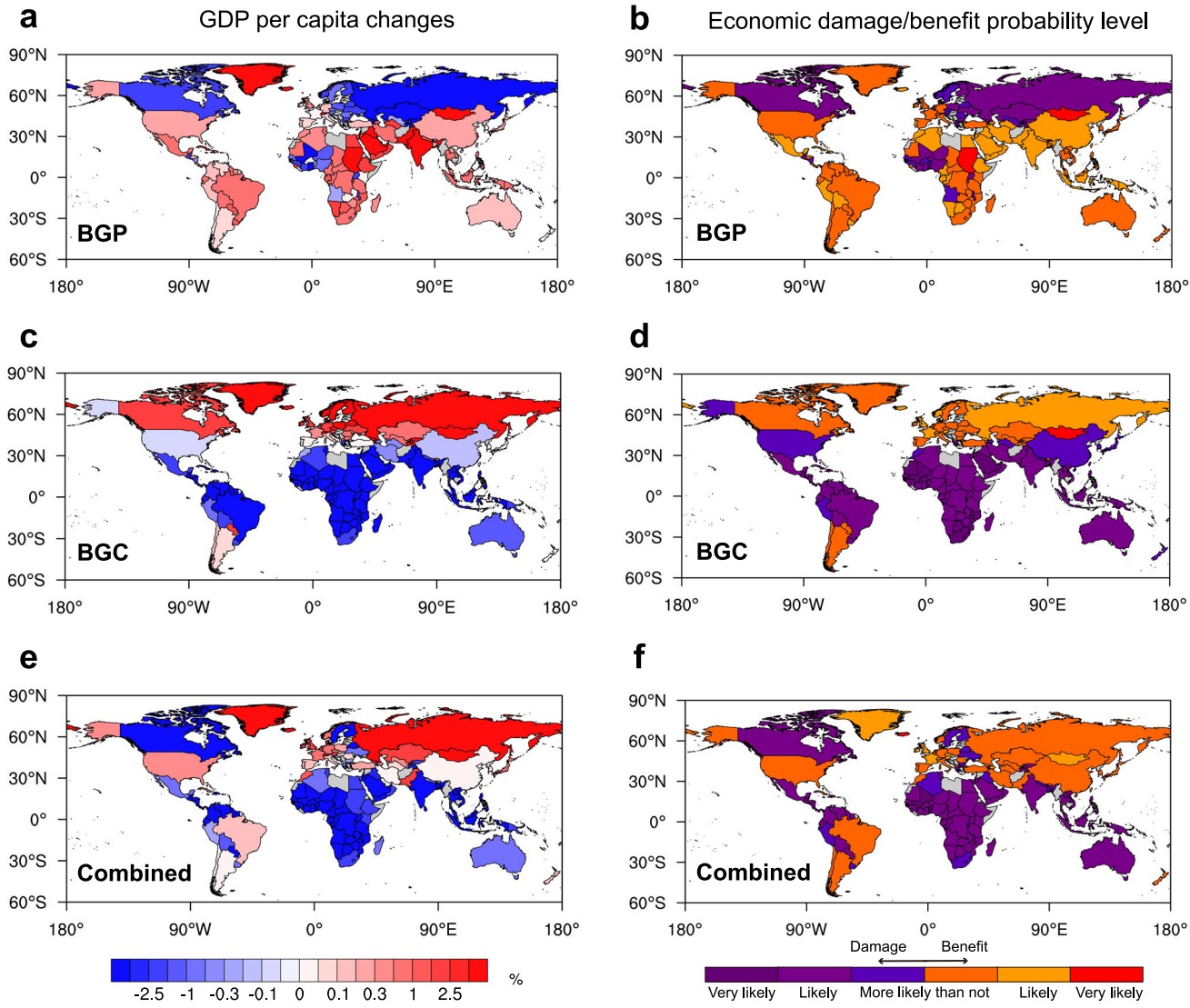

**Fig. 5 Country-level cumulative economic impacts via annual-mean surface air temperature (SAT) changes from 1993 to 2012 due to biogeophysical (BGP) and biogeochemical (BGC) impacts of historical land-use and land-cover change (LULCC). a, c, e** The ensemble median of the relative changes in GDP per capita (%) in 2012 induced by (**a**) BGP, (**c**) BGC, and (**e**) their combined impacts. **b, d, f** The corresponding probability level of the economic damage/benefit according to the IPCC uncertainty guidance[66] for (**b**) BGP, (**d**) BGC, and (**f**) their combined impacts. "Very likely", "Likely", and "More likely than not" indicate that more than 90%, two-thirds, and half of the members agree on the response, respectively. Countries and regions with missing values are shaded in gray.

extratropics (e.g., Canada, the US, and Western European countries) benefit from reduced day-to-day SAT variability there. Quantitatively, the 80:20 and 90:10 ratios of the global economic gap are increased by +9.36% and +2.49%, respectively, due to the combined impact of LULCC. Therefore, in addition to the annual mean SAT impact, LULCC-induced day-to-day SAT variability changes further deteriorate global economic inequality.

We further decompose the impact of annual mean day-to-day SAT variability changes on the annual GDP per capita into economic contributions from day-to-day SAT variability changes in four seasons[25] (see Methods). It is shown that spring dominates the annual economic impacts in many countries (Supplementary Figs. 5 and 6). This is mainly due to greater economic sensitivities to spring day-to-day SAT variability changes compared with the other three seasons[25]. In addition, LULCC impacts on spring day-to-day SAT variability are also prominent (Supplementary Fig. 5a). Overall, the total impact of seasonal day-to-day SAT variability changes on annual GDP per capita is comparable to that calculated by

annual mean day-to-day SAT variability changes with slightly noticeable differences in the USA, Russia, and Finland (Supplementary Fig. 6e, f).

## Discussion

This study investigates individual and combined BGP and BGC impacts of historical LULCC on annual mean SAT by multi-model climate simulations from CMIP6 and estimates economic effects of these SAT changes according to the bootstrapped temperature–growth response function[27]. LULCC results in BGP cooling but BGC-dominated global warming. Thus, there are contrasting influences of BGP and BGC impacts on the economy. Given the dominant role of the BGC impact, historical LULCC leads to increased global economic inequality. The conclusions remain unchanged using different SAT reanalysis (Supplementary Figs. 7–9) and temperature–growth response functions (Supplementary Tables 2–4) (see Methods). We also explore the impact of LULCC-induced day-to-day SAT variability changes on

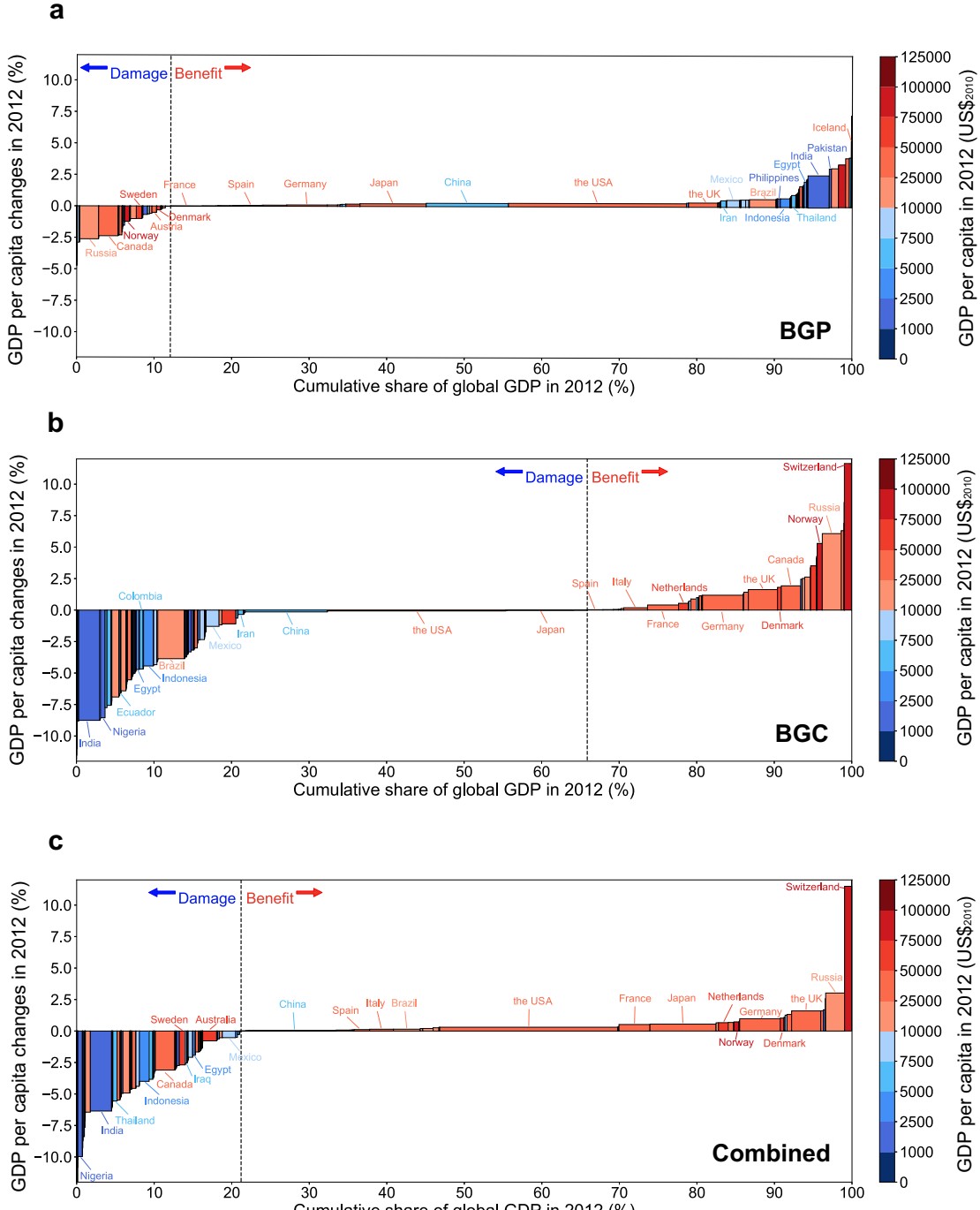

**Fig. 6 The biogeophysical (BGP) and biogeochemical (BGC) impacts of historical land-use and land-cover change on economies in countries of different economic conditions via annual-mean surface air temperature (SAT) changes.** Countries are sorted by their relative changes in GDP per capita (%) in 2012 induced by (**a**) BGP, (**b**) BGC, and (**c**) their combined impacts, from damages on the left to benefits on the right, with bars shaded by their corresponding GDP per capita in 2012 (US$_{2010}$). Blue/red color is split at the global GDP per capita in 2012 (approximately 10,000 US$_{2010}$)[39]. Countries with GDP per capita greater than 10,000 US$_{2010}$ are grouped into economically advanced countries (in red). Otherwise, they are grouped into economically disadvantaged countries (in blue). The x-axis indicates the share of global GDP for each country.

economies and find that global economic inequality is further enlarged with a major contribution from spring.

In addition to $CO_2$, historical LULCC also emits other greenhouse gases, such as methane ($CH_4$), nitrous oxide ($N_2O$), and ozone ($O_3$)-producing compounds[20,53]. A recent study quantified the climate forcing of increased concentrations of $CO_2$ and non-$CO_2$ greenhouse gases due to historical LULCC[20]. They found that $CO_2$ is the greatest LULCC forcing agent and that the

radiative forcing of other non-$CO_2$ greenhouse gases further enhances climate warming. This implies that the magnitude of the increased global economic inequality in this study will be greater with additional warming impacts from other non-$CO_2$ greenhouse gases. There are some uncertainties in historical land-use forcing. To present the upper and lower bounds of the uncertainties in historical land-use forcing, LUH2 developed two extreme historical land-use reconstructions ("high" and "low"

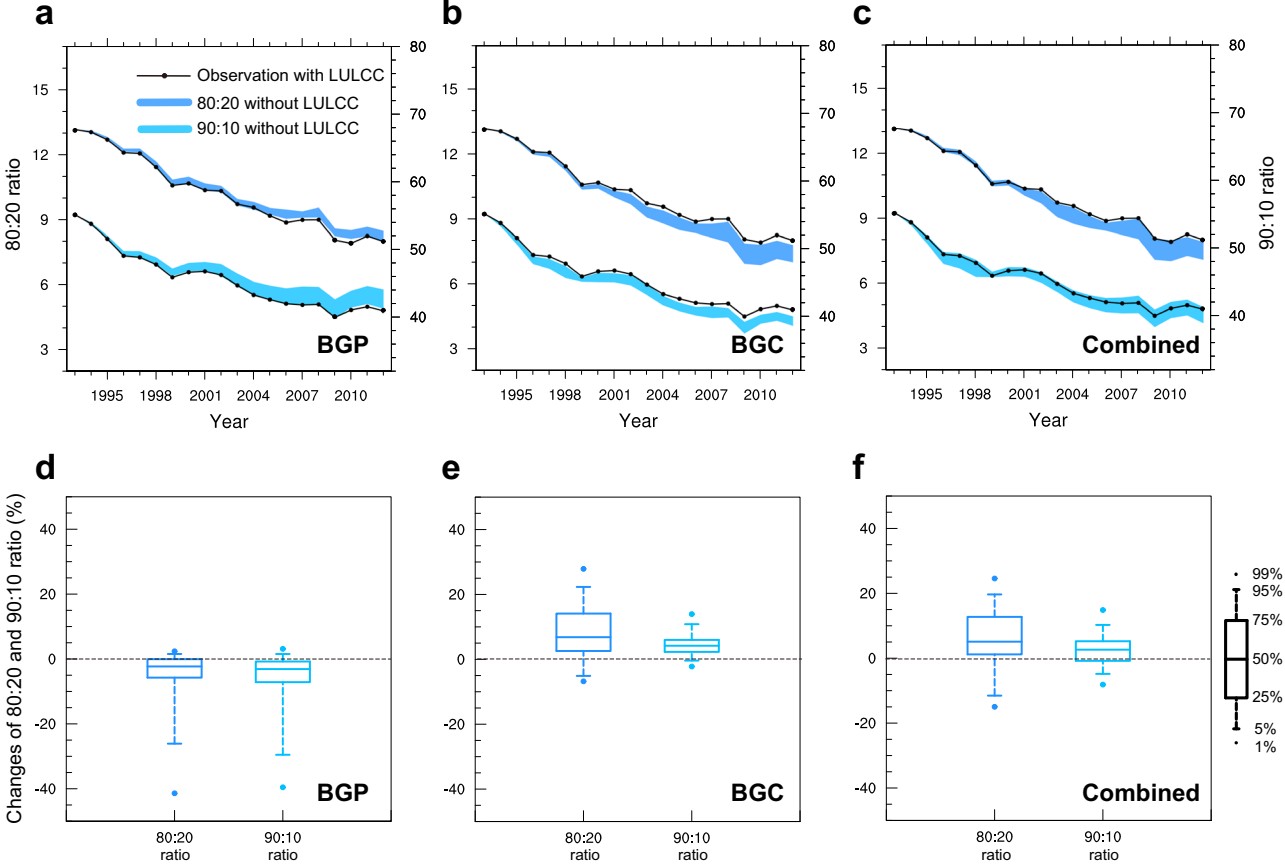

**Fig. 7 The biogeophysical (BGP) and biogeochemical (BGC) impacts of historical land-use and land-cover change (LULCC) on global economic inequality via annual-mean surface air temperature (SAT) changes. a–c** Time series of the 25–75th percentile range of 80:20 (dark blue shading) and 90:10 (light blue shading) ratios of the population-weighted percentiles of GDP per capita in the counterfactual world without (**a**) BGP, (**b**) BGC, and (**c**) their combined impacts, both with corresponding observations in the factual world (black dotted line), from 1993 to 2012. **d–f** Relative changes (%) in 80:20 (dark blue box) and 90:10 (light blue box) ratios in 2012 induced by (**d**) BGP, (**e**) BGC, and (**f**) their combined impacts. The black box on the right shows the distribution of percentiles of box-and-whiskers. There are 16,000 members to estimate the BGP impact and 9,000 members to estimate the BGC impact and their combination (see Methods).

land-use) in terms of the cumulative wood harvested and total area of forests removed[4,43]. The land-use uncertainties may affect the magnitude of LULCC-induced SAT changes. Nonetheless, the spatial patterns of the changes are kept unchanged[9–18].

Given the socioeconomic data of most countries dating back to just recent decades[39], we choose to study a shorter time slice (from 1993 to 2012) to cover as many countries as possible. A longer study period yields a larger magnitude of economic impacts, as LULCC effects on annual economic growth accumulate over time[27–29]. This implies that the impact of LULCC on the global economy since the Industrial Revolution is greater than the impact accumulated just over the two decades considered here.

With all the influencing factors (e.g., trade globalization), global economic inequality over the past decades has been mitigated[54,55]. Decomposing the declining economic inequality into contributions from between- and within-country, it has been found that the economic gap between developed and developing countries accounts for four-fifths of the declining trend of global inequality[54]. As the trend of global inequality is dominated by the between-country economic gap, the analysis is performed at the country level. The climate impacts of historical LULCC negatively impact the decreasing trend of country-level global economic inequality. Some previous analyses on economic inequality were based on household income or consumption data according to national statistical surveys[39,54,55], showing that economically

disadvantaged people are losing more relative to economically advanced people when exposed to extreme heat[56]. Therefore, global warming induced by the combined effects of historical LULCC also deteriorates economic inequality at the individual or household level[56], in addition to the country level.

In comparison with the within-country regional level, the country-level ability to resist risks of climate change is greater because adaptation is likely to be coordinated among different regions at the country level[25]. Attention should also be paid to the within-country regional economic impact under climate change because of higher regional climate vulnerability[25]. For instance, LULCC has significant impacts on precipitation as well[12,33]. Nonetheless, no significant effects of precipitation on country-level GDP per capita were found[25–27], but some analyses show that precipitation impacts on within-country regional economic production are noticeable[57–59]. This implies the necessity of within-country regional analyses in countries covering large areas, particularly for analyzing the economic impacts imposed by climate factors that are strongly spatially heterogeneous. However, a global within-country analysis is challenging owing to insufficient global spatiotemporal subnational socioeconomic data[25,39]. To overcome this issue as much as possible, we aggregate temperature to the country level by population weighting in the country. This provides some indications of the within-country regional economic gap; temperature changes in sparsely

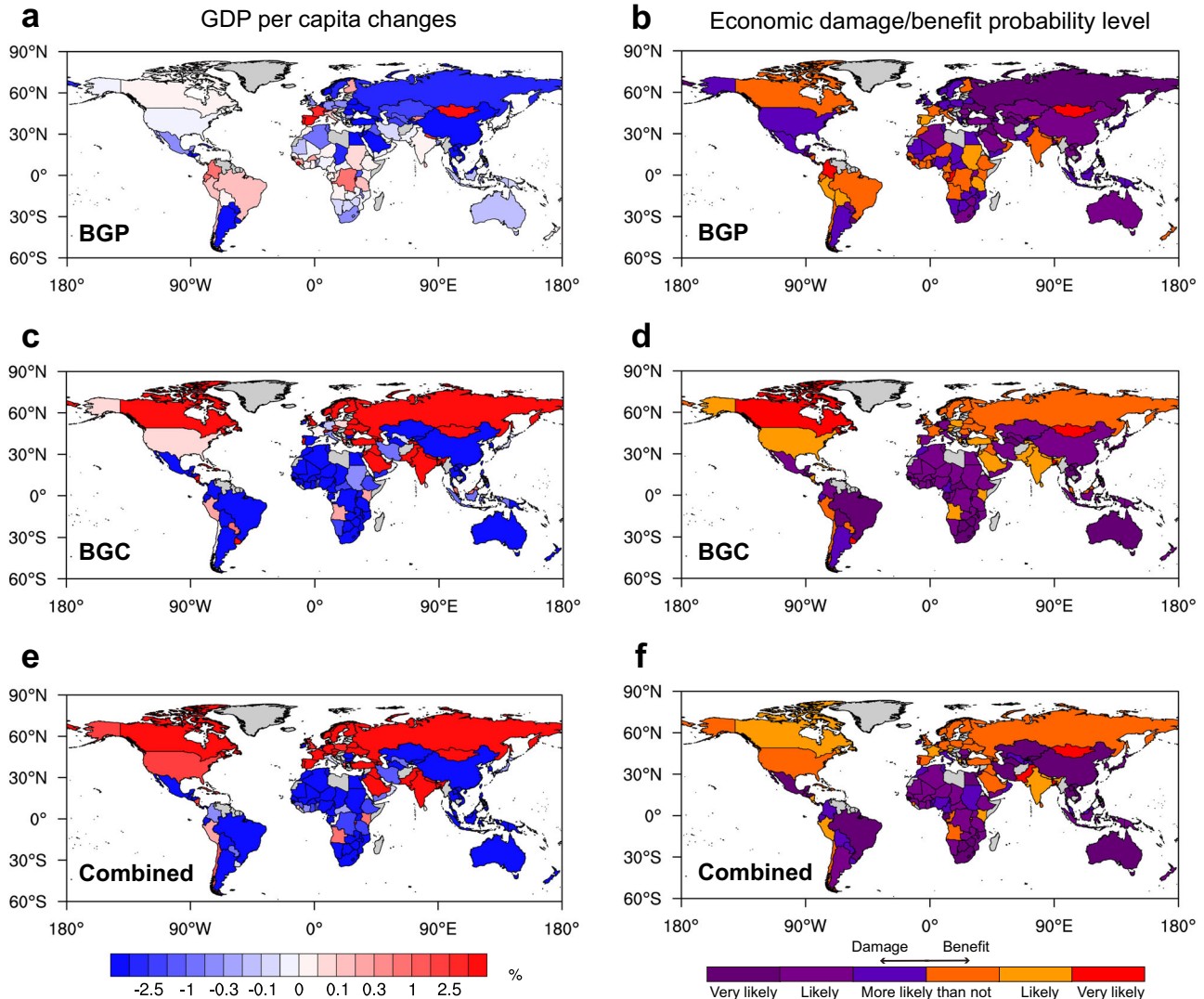

**Fig. 8 Country-level cumulative economic impacts via annual-mean day-to-day surface air temperature (SAT) variability changes from 1993 to 2012 due to biogeophysical (BGP) and biogeochemical (BGC) impacts of historical land-use and land-cover change (LULCC).** The ensemble median of the relative changes in GDP per capita (%) in 2012 induced by (**a**) BGP, (**c**) BGC, and (**e**) their combined impacts. The corresponding probability level of the economic damage/benefit according to the IPCC uncertainty guidance[66] for (**b**) BGP, (**d**) BGC, and (**f**) their combined impacts. "Very likely", "Likely", and "More likely than not" indicate that more than 90%, two-thirds, and half of the members agree on the response, respectively. Countries and regions with missing values are shaded in gray.

populated areas with less economic activities contribute to less economic impacts in the country.

Despite short-term regional economic growth brought by land-use activities with gains of agricultural/industry commodities, this study reveals the potential declines in global economic equitable and sustainable development due to climate feedback of LULCC. The extent to which society should limit its impact on the eco-system to prevent long-term economic damages with short-term benefits poses a difficult challenge for policymakers. The solution hinges on estimating overlooked potential long-term economic damages behind short-term economic growth. A sound under-standing of socio-climatic interactions is vital for assessing where the optimal balance of global economic development and envir-onmental protection lies. Historical warming induced by anthropogenic activities has caused economies of developing countries to be more vulnerable to climate change. In the warmer future, economies will be damaged more seriously from warming. Therefore, quantification of potential economic damage/benefit of future LULCC of different scenario trajectories (e.g., deforestation

and afforestation) is central to planning appropriate land-use policies for the sustainable development of human society[4,43].

## Methods

**Historical land-use data.** Land-use forcing for the "historical" experiment (from 1850 to 2014) in CMIP6 is based on the Land-Use Harmonization 2 (LUH2) project[4,43,46]. The LUH2 dataset represents five broad land-use categories: primary vegetation, secondary vegetation, grazing land, cropland, and urban land. Primary and secondary vegetation are both natural vegetation (either forest or non-forest), but primary vegetation has not been disturbed by humans, while secondary vegetation is re-growing vegetation recovering from previous human disturbance. The dataset also includes several land-use management layers, such as wood harvest (disaggregated into fuelwood and industrial-use roundwood), irrigation, and industrial nitrogen fertilizer usage.

**CMIP6 experiments.** Multi-model climate simulations from CMIP6 are used to estimate the BGP and BGC impacts of historical LULCC on SAT[42]. Two concentration-driven climate simulation experiments, both conducted from 1850 to 2014, are compared and analyzed. One is the "historical" experiment, which is driven by historical evolving natural (e.g., solar, orbital, and volcanic) and anthropogenic (e.g., land-use, greenhouse gases, and aerosol) forcings[42]. The other is the "hist-noLu" experiment, which is identical to the former, except all land-use

and land-cover are fixed at the 1850 level during the historical simulation from 1850 to 2014[43]. The modeled SAT differences between the two experiments only isolate the BGP impact of LULCC because of the same $CO_2$ concentrations prescribed in the two concentration-driven experiments without climate feedback of LULCC-induced $CO_2$ emissions (i.e., the BGC impact)[43]. Despite the same atmospheric $CO_2$ concentrations, carbon cycle processes (e.g., the land-atmosphere $CO_2$ flux) can be simulated according to the land surface and climate conditions in the two experiments[43]. To derive the BGC effect of historical LULCC, the CMIP6 "1pctCO2" experiment is used. The "1pctCO2" experiment, where $CO_2$ concentration is increased gradually at a rate of 1% per year since 1850, is used to isolate the SAT response of accumulated LULCC-induced increases in $CO_2$ concentrations since 1850 (see estimates of LULCC impacts on SAT below)[42,60,61].

Supplementary Table 1 summarizes different ensemble sizes used for estimating annual mean SAT and day-to-day SAT variability changes induced by the BGP and BGC impacts of LULCC. Due to data availability, a total of 16 members from 8 participating global climate models (GCMs) are selected to estimate the BGP impact of historical LULCC on the annual mean SAT. For the estimate of the BGC impact on annual mean SAT, the ensemble size is reduced to 9 members from 4 participating GCMs due to other GCMs without explicitly treating the carbon cycle processes in the "historical" and "hist-noLu" experiments. Therefore, when combining the BGP and BGC impacts on SAT, 9 members are used. Note that the reduced ensemble size does not change the results of the BGP impact of historical LULCC (Supplementary Fig. 10). For the estimates of the BGC and BGP impacts on day-to-day SAT variability, the ensemble size is reduced to five members from three participating GCMs due to other GCMs without daily SAT outputs in the "hist-noLu" and "1pctCO2" experiments. To be consistent, all GCM outputs are interpolated to the same resolution of 2.5° × 1.9°, which is the coarsest grid in the selected GCMs.

**Estimates of LULCC impacts on SAT.** The BGP impact of historical LULCC on SAT is estimated by directly comparing modeled SAT between the "historical" and "hist-noLu" experiments. The BGC impact of LULCC on SAT is estimated by calculating accumulated LULCC-induced increases in $CO_2$ concentrations since 1850[20] and applying them to the "1pctCO2" experiments. Although the "historical" and "hist-noLu" experiments in CMIP6, unlike emission-driven simulations, are concentration-driven simulations with atmospheric $CO_2$ concentrations prescribed, some GCMs still compute the land-atmosphere $CO_2$ fluxes offline accounting for atmospheric feedback[43]. As such, global yearly LULCC-induced $CO_2$ emissions are calculated with the annual differences in the global net land-atmosphere $CO_2$ fluxes between the "historical" and "hist-noLu" experiments[20,43,62]. It accounts for the net exchange of $CO_2$ between the land and atmosphere associated with LULCC (e.g., deforestation and reforestation)[20,63]. It also includes the BGP feedback of LULCC because of the climatic differences between the two experiments due to the BGP impact[20,43,63,64]. $CO_2$ is chemically inert in the atmosphere, but over time, the airborne fraction of emitted $CO_2$ concentration decreases with carbon uptake of ocean and land, the highest airborne fraction for the most recent $CO_2$ emissions. Following prior studies[20,63], we calculate the airborne fraction of the yearly LULCC-induced $CO_2$ emissions since 1850 using a $CO_2$ pulse response function that presents the residual fraction of $CO_2$ over time due to carbon uptake[63]. Annual LULCC-induced $CO_2$ emissions are multiplied by corresponding airborne fractions and then summed over time since 1850 to yield accumulated atmospheric $CO_2$ concentrations at present. Here, the historical LULCC contribution to atmospheric $CO_2$ concentration is approximately 25.3 ppm (the median of ensemble members) from 1850 to 2014, an increase of 9% relative to the atmospheric $CO_2$ concentration in 1850. The SAT responses to the accumulated $CO_2$ concentrations are estimated by the "1pctCO2" experiment, which accounts for different climate sensitivities in each model to the increasing $CO_2$ concentrations and maps the global distribution of corresponding SAT changes[60,61]. For example, given an increase of 9% in 2014 relative to 1850, the corresponding LULCC-induced SAT differences of the BGC impact in 2014 are calculated as the SAT in the 9th year after 1850 minus that in 1850 in the "1pctCO2" experiment. Note that the 1pctCO2 simulation is not in equilibrium for the increasing $CO_2$ concentrations. It is a compromise to estimate the $CO_2$ effect here because there is currently not a set of CMIP6 experiments that allows us to directly evaluate the BGC impact of historical LULCC. The BGP and BGC impacts can be arithmetically summed to obtain their combined impacts[20,44,45,65]. Our derived results of LULCC-induced $CO_2$ emissions and SAT changes are comparable to those of previous studies[17-20,45,47,49,53,65].

To reduce the interannual variability as much as possible, we calculate the 5-year running averages of SAT. For instance, SAT in 2012 refers to the annual mean SAT during 2010–2014. We focus on the impacts of historical LULCC on SAT and economies during the study period from 1993 to 2012, given the socioeconomic data of most countries dating back to just recent decades[39]. Note that land-use and land-cover differences between the "historical" and "hist-noLu" experiments during the study period actually show historical LULCC since 1850 because all land-use and land-cover are maintained at the 1850 level in the "hist-noLu" experiment during the simulation from 1850 to 2014.

**Spatial country-level aggregation.** Grid-cell SAT is aggregated to the country level, weighted by population distribution in the country according to the gridded population data from the Gridded Population of the World version 4 (GPWv4)[41]. The aggregated country-level SAT reflects the representative SAT where the majority of populations and economic production of the country are situated; temperature changes in sparsely populated areas with less economic activities contribute to less economic impacts in the country.

**Factual and counterfactual worlds.** We compare country-level SAT and economic conditions between two worlds: one is the factual world with LULCC, and the other is the counterfactual world, which is identical to the former except without the BGP and BGC impacts of LULCC. Given the existing SAT biases in CMIP6, the SAT differences induced by LULCC ($\triangle T_{LULCC}$) derived from the "historical" and "hist-noLu" experiments are used only. We use reanalysis data to represent the SAT in the factual world ($T_{Obs}$). Applying the simulated LULCC-induced SAT difference to the reanalysis, we obtain the SAT in the counterfactual world without LULCC ($T_{NoLULCC}$):

$$T_{NoLULCC} = T_{Obs} - \triangle T_{LULCC} \qquad (1)$$

This approach of bias correction is called the "delta" method, which has been widely used in previous studies[28]. Similarly, socioeconomic statistical data are used to represent the economic conditions of the factual world. Accordingly, the counterfactual economies without LULCC are estimated using the factual statistics minus the LULCC-induced economic differences with the "delta" method (see estimates of economic impacts of annual mean SAT changes below). Thus, the calculation of SAT and economies in the counterfactual world is constrained by observations.

**SAT reanalysis and socioeconomic statistical data.** Global gridded SAT in the factual world from 1993 to 2012 are derived from the European Center for Medium-Range Weather Forecasts (ECMWF) Reanalysis version 5 (ERA5)[38] and the Modern-Era Retrospective analysis for Research and Applications Version 2 (MERRA2)[40]. The former is used for the central estimate and the latter is used for the sensitivity test. They are interpolated to the 2.5° × 1.9° resolution and then aggregated to the country level. The sensitivity test demonstrates the robust historical LULCC impacts despite the different SAT reanalysis data used (Figs. 5–7 and Supplementary Figs. 7–9). Socioeconomic statistical data from 1993 to 2012, such as GDP per capita, GDP per capita growth, and population, are obtained from the World Bank database[39]. 147 countries are available for analysis.

**Temperature–growth response function.** Recent research assembled historical climatic and economic data (covering 165 countries from 1960 to 2010) and built the empirical relationship of country-level annual mean temperature and GDP per capita growth rate[27,28]. It is referred to as the "temperature–growth response function", as follows:

$$f(T) = \beta_1 T + \beta_2 T^2 \qquad (2)$$

where $T$ is the country-level annual mean SAT (°C), $f(T)$ shows the response of the GDP per capita growth rate to the annual mean SAT, and $\beta_1$ and $\beta_2$ are parameters of the response function ($\beta_1 > 0$; $\beta_2 < 0$). The function shows nonlinear responses of economic growth to changes in annual mean temperature. The statistical relation is in accord with the empirical evidence that in the hottest and coldest areas on the earth, per capita economic productivity is low, and that, when other conditions are equal, economic productivity peaks at some intermediate temperature. This implies that temperature affects macro-economic growth by influencing agricultural yields, energy supply, labor productivity, and human health, as shown in previous studies[21]. As illustrated in Fig. 4a, the growth rate of GDP per capita peaks at the optimal temperature and declines at higher or lower temperatures. As background temperature deviates from the optimum, economic production gradually becomes more sensitive to temperature disturbance.

Our study uses the primary form of the response function for central estimates[27]. It adopts the bootstrapping strategy of sampling by country 1000 times, with no lag of temperature; 1000 members of the response function (1000 sets of parameters) are applied. Thus, a country-level population-weighted SAT change induced by LULCC corresponds to 1000 responses of GDP per capita growth. The set of response functions shows 1000 cases of the "temperature optimum" of economic growth, with a median estimate of 13.12 °C in a 25–75% range of 11.80–14.55 °C. To test the sensitivity to the response function adopted, we also apply several other response functions (all with 1000 members as well) derived from different bootstrapping strategies, lags of temperature, or observations (Supplementary Tables 2–4)[23,26,27]. Overall, the conclusions are kept unchanged regardless of which response function is used.

**Estimates of economic impacts of annual mean SAT changes.** The impact of annual mean SAT changes on the growth rate of GDP per capita is estimated following the bootstrapped temperature–growth response function[27]. Note that the ensemble sizes of the economic impacts are increased by 1000 times relative to those of the climatic impacts because of the 1000 members of the response function used. Therefore, there are 16,000 members to estimate the BGP impact of historical LULCC on the economy and 9000 members to estimate the BGC impact and their combination. Applying country-level factual SAT ($T_{Obs}$) and counterfactual SAT

$(T_{NoLULCC})$ to the bootstrapped temperature–growth response function ($f(T)$), respectively, LULCC-induced changes in the growth rate of GDP per capita ($\triangle Growth_{LULCC}$) can be obtained:

$$\triangle Growth_{LULCC} = f(T_{Obs}) - f(T_{NoLULCC}) \quad (3)$$

$\triangle Growth_{LULCC}$ is further applied to factual statistics ($Growth_{Obs}$) to yield the growth rate of GDP per capita in the counterfactual world ($Growth_{NoLULCC}$) using the "delta" method:

$$Growth_{NoLULCC} = Growth_{Obs} - \triangle Growth_{LULCC} \quad (4)$$

For the GDP per capita in the counterfactual world ($GDPcap_{NoLULCC}$), we first initialize it with the corresponding factual statistics ($GDPcap_{Obs}$) in the starting year of the study period (i.e., $y = 1993$) for each country:

$$GDPcap_{NoLULCC}(1993) = GDPcap_{Obs}(1993) \quad (5)$$

After the starting year ($y > 1993$), the GDP per capita in the counterfactual world ($GDPcap_{NoLULCC}(y)$) is determined by the current-year counterfactual growth rate ($Growth_{NoLULCC}(y)$) multiplied by last year counterfactual GDP per capita ($GDPcap_{NoLULCC}(y-1)$):

$$GDPcap_{NoLULCC}(y) = GDPcap_{NoLULCC}(y-1) + (GDPcap_{NoLULCC}(y-1) \\ * Growth_{NoLULCC}(y)) \quad (6)$$

Following the process through the study period from 1993 to 2012[28], we finally create the counterfactual GDP per capita and its growth rate for each country of each year during 1993–2012, which can be compared with the observed factual world with LULCC, to study LULCC impacts on the global economy. The relative change in GDP per capita induced by LULCC ($\triangle GDPcap_{LULCC}$) is calculated as the GDP per capita difference between the factual and counterfactual worlds relative to the counterfactual world:

$$\triangle GDPcap_{LULCC} = \left[ \frac{GDPcap_{Obs} - GDPcap_{NoLULCC}}{GDPcap_{NoLULCC}} \right] \times 100\% \quad (7)$$

The temperature–growth response function shows that annual temperature fluctuations influence annual GDP per capita growth rates, so impacts on GDP per capita accumulate over time. For instance, if LULCC-induced warming reduces the GDP per capita growth rate annually, the damage to GDP per capita will increase with time. We study LULCC impacts from 1993 to 2012 and finally focus on the cumulative economic impacts in the last year of the study period (i.e., 2012).

**Estimates of economic impacts of day-to-day SAT variability changes**. In addition to annual mean SAT, we also calculate day-to-day SAT variability, which reflects the magnitude of daily temperature fluctuations; greater temperature variability implies stronger heat or/and cold extremes[48]. The annual mean day-to-day SAT variability is calculated as the intra-monthly standard deviation of daily SAT averaged across 12 months of a year[25]. The measure of day-to-day SAT variability is seasonally adjusted because economic agents are already adapted to the seasonal cycle[25]. Both heat and cold extremes are involved in day-to-day SAT variability. A recent study has documented a damage function describing country-level responses of the GDP per capita growth rate to every extra degree Celsius of the day-to-day SAT variability[25]:

$$\triangle Growth_{LULCC} = \alpha \times \triangle TVAR_{LULCC} \quad (8)$$

where $\triangle TVAR_{LULCC}$ is the LULCC-induced country-level day-to-day SAT variability change and $\triangle Growth_{LULCC}$ is the associated change in the growth rate of GDP per capita. $\alpha$ is a negative parameter denoting the decrease in the growth rate of GDP per capita from an extra degree Celsius of day-to-day SAT variability (greater $|\alpha|$ implies greater damage). The parameter is dependent on the country-level seasonal temperature difference. The magnitude of $\alpha$ is greater in countries with smaller seasonal variations (e.g., tropical countries). Using the damage response function and then following the same process as the estimates of economic impacts of annual mean SAT changes, the economic impacts of day-to-day SAT variability changes can be derived with country-level day-to-day SAT variability changes induced by LULCC ($\triangle TVAR_{LULCC}$) calculated from CMIP6 simulations.

In addition to the annual mean seasonally adjusted day-to-day SAT variability (averaged across 12 months), we also calculate seasonal day-to-day SAT variability averaged across 3 months of the season for boreal spring (March, April, May), summer (June, July, August), autumn (September, October, November), and winter (December, January, February). Then, the impacts of these seasonal day-to-day SAT variability changes induced by LULCC on annual GDP per capita are investigated according to a season-specific damage function developed by Ref.[25]. These seasonal influences show their respective economic contributions to annual GDP per capita changes and can be summed to reflect annual cumulative economic impacts with the seasonal cycle. For a given day-to-day SAT variability, its economic impact varies in different seasons[25]. Compared with other seasons, spring shows greater economic damage in response to increased day-to-day SAT variability[25]. Economic growth in winter is insensitive to day-to-day SAT variability changes because economic agents may be sheltered from extreme weather effects by reducing outdoor activities[25].

**Quantification of changes in global economic inequality**. To quantitatively estimate the impacts of LULCC on global economic inequality, we use common measures of economic inequality: the 80:20 and 90:10 ratios of the population-weighted percentile of GDP per capita[52]. Both the 80:20 and 90:10 ratios reflect the global economic gap between the top economically advanced country and the bottom economically disadvantaged country; the greater ratios are, the wider the gap. Specifically, the 80th percentile of population-weighted GDP per capita is calculated as the GDP per capita, for which the total population of countries with lower GDP per capita occupies 80% of the total population of countries. The calculation of other percentiles of population-weighted GDP per capita is similar to that of the 80th percentile. The 80:20 ratio reflects the global economic inequality between the top 20% economically advanced country and the bottom 20% economically disadvantaged country. Similarly, the 90:10 ratio shows the global economic gap between the top 10% economically advanced country and the bottom 10% economically disadvantaged country. The results are not sensitive to the different economic inequality indicators used (Fig. 7 and Supplementary Fig. 9).

**Significance test**. The median and 25–75th percentile range of ensemble members are presented for analysis. Changes are considered to be significant if more than two-thirds of members agree on the sign of response. The economic damage/benefit probability is calculated as the percentage of members that show decreases/increases in GDP per capita. It is further aggregated into the probability levels of "Very likely", "Likely", and "More likely than not" according to the IPCC uncertainty guidance[66]. The probability levels of "Very likely", "Likely", and "More likely than not" indicate that more than 90%, two-thirds, and half of the members agree on the economic response, respectively.

**Reporting summary**. Further information on research design is available in the Nature Research Reporting Summary linked to this article.

## Data availability

Data that support the findings are publicly available. The LUH2 land-use data are available at https://luh.umd.edu/index.shtml. The CMIP6 model outputs are accessible via the website https://esgf-node.llnl.gov/search/cmip6/. The ERA5 surface air temperature data are available at https://cds.climate.copernicus.eu/cdsapp#!/dataset/reanalysis-era5-single-levels-monthly-means?tab=overview. The MERRA2 surface air temperature data are available at https://disc.gsfc.nasa.gov/. The GPWv4 gridded population data can be accessed via https://sedac.ciesin.columbia.edu/data/collection/gpw-v4. The socioeconomic statistical data from the World Bank are publicly available at https://databank.worldbank.org/source/world-development-indicators. The temperature–growth response function is based on https://purl.stanford.edu/vn535jm8926. The responses of economic growth to day-to-day temperature variability changes are based on https://zenodo.org/record/4323163#.YixvDIlBw2w. The data for the counterfactual world without LULCC generated in this study are deposited in https://zenodo.org/record/6349650#.Yi1XQo9Bw2w.

## Code availability

Data were analyzed with publicly available software: NCAR Command Language, Climate Data Operators (CDO), and Python. All the scripts are available upon request.

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

## Acknowledgements

Y.W. is supported by the National Key Research and Development Program of China grant 2017YFA0604000 and the National Natural Science Foundation of China grant 41975126. L.Y. is supported by Tsinghua University Initiative Scientific Research

Program (2021Z11GHX002) and the National Key Scientific and Technological Infrastructure project "Earth System Science Numerical Simulator Facility" (EarthLab). We thank Wei Li at Tsinghua University for useful discussions.

## Author contributions

Y.W. conceived and supervised the study. Y.W. and S.L. designed and conducted the analysis. S.L. and Y.W. wrote the paper. G.J.Z., B.W., L.W., and L.Y. discussed the results and edited the manuscript.

## Competing interests

The authors declare no competing interests.
