## [Peer Review File · Nature Communications]

REVIEWER COMMENTS, first round

Reviewer #1 (Remarks to the Author):

Re "Global Economic Inequality Reduced by Historical Land-Use and Land-Cover Change"

Major comments:

This study investigated the historical land-use and land-cover changes (LULCC) and their impacts on the global economic inequality. The LULCC-induced cooling harmed wealthier economies in colder regions but benefitted poorer economies in hotter regions and alleviated the global economic inequality by 4.60%. The writing is of high quality, the methodology and the analysis are generally sound, but the introduction needs more works: 1) the importance of the LULCC-induced temperature; 2) the connection of LULCC-induced temperature and economy. It seems that the author did not consider the effects of some extreme temperature or temperature seasonal response on the economy, if so, please add some explanation why they did not been included. In the Method, please add some explanation of the uncertainties of LULCC data and ERA5. Overall, I think this manuscript could be considered for publication after a major revision.

Specific comments:

Line 18 "warming has been shown to harm economies in warm climates but provide benefits in cold climates." Which are specific aspects in economies? And why?

Line 21 "unlike anthropogenic aerosols, the climate effects of LULCC differ in latitudes and disturbed land surface types." What is effect of anthropogenic aerosols?

Line 22 "we presented historical LULCC-induced temperature changes by multi-model climate simulations from the Coupled Model" please explain what is "LULCC-induced temperature" and which simulations?

Lines 25-26 "Historical LULCC resulted in strong cooling of up to..." please define a specific period in historical; "... some cooling/warming over tropics... ..." please make this clear.

In the Introduction, please add some previous studies about LULCC-induced temperature changes and say why "LULCC-induced temperature" is important.

Lines 253-256 "With all forcing agents can be attributed to historical LULCC." This sentence should be mentioned in the introduction.

Line 103 In fig1, is there no urban land fraction showing? The period is from 1850-2014 in figs 1-2, which should be changed into the period of 1991-2010 for economy responses as in fig 3?

Lines 130-131 "Greenland and central Africa experienced some warming induced by LULCC. " What LULCC in Greenland and central Africa? How this LULCC induced the warming with bio-geophysical-chemical processes?

Lines 143-145 "we found that countries of low latitude were above the temperature optimum, while those of high latitude were below the optimum." Clarify low/high latitude.

In the results section: the effects of the extreme temperature and seasonal cycles on the economy were considered in different countries? If this included, will change the conclusions you found now?

Line 152, why were India and Canada selected?

Lines 175-178 "For example, owing to the annual small warming impacts from the LULCC, GDP per capita of Greenland and Iceland in cool climates were estimated to gain in 2010, while that of warm countries over the southwest of Sahara were estimated to lose." Why does this happen?

Line 179 how define the poor (rich) country?

In the uncertainties section, some uncertainties of the land-use and land-cover data and ERA5 should be included.

Line 296 "the high-resolution requirement" how was the high-resolution defined?

Reviewer #2 (Remarks to the Author):

In their paper "Global Economic Inequality Reduced by Historical Land-Use and Land-Cover Change", the authors use a relationship between annual mean temperature and economic success to attribute changes in countries' GDP to historical land use – or, more specifically, to the strictly

biophysical impact of historical land use on annual mean temperatures. They use two sets of CMIP6 model experiments – a “regular” historical simulation and a simulation that does not have historical land use (but has prescribed CO₂ concentrations, thus only captures the biophysical impacts of historical land use), as well as observations of historical climate, to determine how historical land use has impacted temperatures. They then apply their temperature-economics relationship to this temperature change to convert the temperature change to an economic change. They find that biophysical effect of historical land use has reduced global economic inequality, as the biophysical effects of historical land use have generally been to cool, which would benefit lower income countries in warm regions, and be detrimental to countries in cooler regions (more of which are higher income).

While I have not methodological problem with the study, I do have rather fundamental concerns about the conclusions the authors draw which may preclude the work from publication in this journal. The authors are aware of these issues – they mention most of them in their discussion at the end of the manuscript – but do not demonstrate in any way that these factors don’t change their conclusions. It is my feeling that these major limitations to their approach need to be addressed systematically in some way prior to publication, or the authors need to greatly reduce the strength of their conclusions.

Specifically, my concerns are that:

- Land use and land cover change (LULCC) have historically had *as large* biogeochemical effects (i.e., impacts on the carbon cycle) as biogeophysical effects (the latter of which are explored here), globally averaged. However, these signals historically have been of opposite sign, largely cancelling out in the global mean. The authors conclude that LULCC has reduced global economic inequality, but they need to quantify or at least estimate how much the biogeochemical effect of LULCC would have changed economic inequality. Have the biogeochemical effects of LULCC increased economic inequality? Moreover, the biogeophysical and biogeochemical effects don’t operate on the same spatial scale; the carbon cycle impacts of LULCC are global, while the biogeophysical effects are mostly local/regional, so since economic inequality is also spatially heterogeneous, it is possible the authors’ conclusion still holds, but it needs to be demonstrated!
- LULCC has a much stronger impact on temperatures and the water/energy cycles on seasonal timescales, rather than annual mean timescales. In addition, LULCC would likely have had a large impact on changes in extremes, and changes in extremes would, I would expect anyhow, have had greater economic impact than changes in annual mean temperatures. However, the authors’ approach of using annual mean temperature to determine economic benefit would completely miss this, and I would expect this to be the larger part of the signal. How can the authors be confident their historical annual mean T / annual mean GDP relationship can be applied to LULCC induced perturbations of annual mean T?

I outline some more specific comments below, but my main concerns are that the authors need to at least estimate the biogeochemical effect of LULCC on GDP in order to make a claim of how LULCC has actually impacted GDP, and that the annual mean temperature effect of LULCC is likely weaker than the seasonal effect, and the impact on economics would likely come from having e.g., a cooler growing season, or a more extreme heat wave, etc.

Specific comments:

- The authors frequently mention the temperature/GDP relationship, but never show the response curve. Since this curve is central to the paper, it should be shown in the paper.
- The title, and lines 26-28 of the abstract, are too strong a claim for what the results show – specifically the paper is only capturing the annual mean biogeophysical effects of LULCC, which are only a fraction of the actual LULCC impact on climate
- Figure 1: I know total area of urban land is small compared to the other land cover types in this figure, but the fact that it shows up on the legend but isn’t visible at all in the plot is a little weird. I’m not sure what the authors should do about this, but one option would be to put write the total

area at both the start and end of the time series in the legend next to each colour, or right on the plot, so the reader can see "ah, this tiny number for urban area, but the increase over the time considered is really big!" (Even though they can't see the line on the figure).

- Line 112: "America" – either change to "United States of America" if you mean the country, or (from the figure, it looks like much more than just the USA), "North America" / "from southern Canada down through Central America" / "The Americas" if you also want to include the changes in South America as part of this sentence, though the authors specifically call out the Amazon.

- Line 145: "America" -> "USA" or "United States of America", "French" -> "France"

- Line 147: I find this usage of brackets, here and several other places in the paper, really difficult to read – and I'm not alone! See for example:

<https://eos.org/opinions/parentheses-are-not-for-references-and-clarification-saving-space>
This is a stylistic comment though, and the authors can choose to ignore it. But it does make the paper harder to read.

- Line 150: I don't think this is verified with observational constraints, or I'm missing something. It is somewhat impossible to verify the main idea in this study with observational constraints – rather, what is shown in this paragraph is just applying the temperature/GDP relationship to observed, rather than modelled, temperatures.

- Figure 5: It would be helpful to give subplots titles. Also, the blue/red shading of bars is rather confusing. Is the colour split at the global median per capita GDP? if so, state. If not, and the split is arbitrary, consider using a continuous (rather than diverging) colour bar

- Line 186-188: it looks, from this figure, like a lot of wealthy countries also benefited – the area under the combined USA/Germany red bar is quite large (hard to tell from this figure if it is larger than the negative red area in the Canada/Russia bars).

- Figure 6: needs a legend, and a more descriptive caption - what are the two different box/whiskers in panel b? I assume one is the change with and one without land use, but add a legend to panel b also. (It is confusing, as the colours match (a), but the line styles don't, so it isn't clear if (a)'s legend also applies to (b)).

- As mentioned above, the temperature/GDP response function needs to be shown somewhere in the paper.

- Line 229: "It is because that a within" – grammar?

- Line 232: I'm glad the authors discuss the limitations of averaging climate (and GDP) over large countries.

- Line 234: the authors say here that they weight the climate response by population density within each country, but I can't find a description of how this was done in the methods.

- Line 360: I assume the bootstrapped temperature-growth functions are not capturing different sensitivities to seasonality and extremes, so what extra information are they giving you?

- The authors do, in the last few paragraphs of the paper, mention both the carbon cycle effects of LULCC and the seasonal impact of LULCC, but they don't attempt to estimate or quantify these factors, which seems critical to actually make any claims about the historical LULCC impact on economic inequality (see main concerns at the start of my comments).

Reviewer #3 (Remarks to the Author):

This manuscript presents results from a study that considers climate impacts from historical land-use change over the period 1991-2010 and the resulting implications for the economic well-being

of different countries in the world. They specifically consider the biogeophysical impacts of land-use change on annual mean surface air temperatures and then use a temperature-growth response function to estimate whether per capita GDP in each country is likely to increase or decrease as a result of that change in temperature. By comparing all countries across the world the authors determine that historical land-use is likely to have led to a reduction in the global income gap between the 80th and 20th percentiles of the global population. This is a very interesting, novel, and thought-provoking study. The methodology is sound and appears to be reproducible. Supporting evidence is provided for all conclusions and it is well-written. It is worthy of publication, subject to consideration of a few comments and suggestions, as detailed below:

- LUH2 is a land-use product not a land cover product. The authors should be careful not to refer to "land cover" when describing the LUH2 historical land-use products in the Methods section.
- Line 276: secondary vegetation is "re-growing" vegetation, not "regrown".
- The term "America" as a country name should be replaced with "USA" throughout the manuscript.
- In the section beginning on line 102 entitled "Historical LULCC since the Industrial Revolution", and in Figures 1 & 2, the presented results appear to be from Hurtt et al. 2020 (reference number 4 in the reference list). However, although Hurtt et al. 2020 is cited early in the Introduction, it is not cited at all in this section or in the captions for Figures 1 & 2. In addition to citing Hurtt et al. 2020 in this section, the LUH2 dataset itself should also be cited since it is being used extensively here and has its own DOI.
- The analysis is based on several simplifying assumptions that allow the study to consider (in isolation from other factors) a specific response to a specific climate forcing. This is a very useful approach, and (as the authors point out) it facilitates understanding of the mechanisms of economic change in response to land-use change, and could potentially guide future land-use policies. On the other hand, it is also useful to place this study into the broader context by also discussing some of the factors not directly considered here. For example, what are impacts of land-use change on other climatic variables such as precipitation and what is the response of economies to those changes? Also, should we also consider the more direct economic impacts of land-use change (such as loss of ecosystem services, or the gain of agricultural commodities)? The authors do mention these simplifying assumptions up front, but some additional discussion in the "Implications and Uncertainties" section towards the end of the manuscript could be helpful.
- Change in per capita gross domestic product is the measure of economic impact chosen for use in this study. However, it provides an average estimate across all citizens within a country, and in many nations there is often a wide disparity between the wealthiest citizens and the poorest. As the authors do point out, subnational estimates of per capita GDP are difficult to obtain, but I am curious whether there are estimates of the spread of income within countries that could be used to estimate within-country economic inequalities and provide better estimates of the resulting global economic inequalities. Or are there measures of economic activity other than GDP that could be used here? Perhaps this is something that could be touched upon in the Discussion section as well.

Reply to the comments by Reviewer #1

We thank the reviewer for the comments and suggestions to improve our manuscript. Below are our point-by-point responses to these comments. The reviewer's comments are in italics, our responses are in normal font, and manuscript revisions are in blue.

Major comments:

This study investigated the historical land-use and land-cover changes (LULCC) and their impacts on the global economic inequality. The LULCC-induced cooling harmed wealthier economies in colder regions but benefitted poorer economies in hotter regions and alleviated the global economic inequality by 4.60%. The writing is of high quality, the methodology and the analysis are generally sound, but the introduction needs more works: 1) the importance of the LULCC-induced temperature; 2) the connection of LULCC-induced temperature and economy. It seems that the author did not consider the effects of some extreme temperature or temperature seasonal response on the economy, if so, please add some explanation why they did not been included. In the Method, please add some explanation of the uncertainties of LULCC data and ERA5. Overall, I think this manuscript could be considered for publication after a major revision.

Reply: Thanks for the reviewer's positive and valuable comments. The importance of the LULCC-induced temperature change has been provided in Lines 41–55 in the revision:

Historical LULCC has also been recognized as a key driver of anthropogenic climate change, according to the Fifth Assessment Report of the Intergovernmental Panel on Climate Change (IPCC)^{9–10} because it regulates the exchanges of momentum, energy, water, and carbon with the atmosphere through biogeophysical (BGP) and biogeochemical (BGC) processes^{11–16}. Global net BGC impacts of LULCC have contributed to approximately 25% of the historical increase in CO₂ emissions since the Industrial Revolution¹⁰. Owing to the spatial heterogeneity of the land surface, the BGP impact of LULCC on temperature spatially is not as homogeneous as warming induced by greenhouse gases and cooling resulting from aerosols⁹. It varies in latitudes and disturbed land surface types^{17–18}. For instance, the BGP impact of deforestation leads to warming in the tropics by reducing evapotranspiration but cooling in the extratropics because of increased surface albedo^{17–19}. In general, the BGP impact of global LULCC contributes to global radiative forcing of -0.15 (-0.25 to -0.05) W/m² relative to the pre-industrial level, masking some of the warming induced by greenhouse gases⁹. When BGP and BGC processes are considered together, up to 40% ($\pm 16\%$) of the present-day global anthropogenic warming can be attributed to historical LULCC²⁰.

The connection of LULCC-induced temperature change and associated economic impact has been discussed in Lines 80–85 in the revision:

It has been documented that long-term economic sustainable development is limited due to substantial disruption to the ecosystem from historical LULCC despite short-term regional economic growth with the gains of agricultural/industrial commodities^{34–37}. Nonetheless, the economic impact of historical LULCC through climate feedback is still unclear. This is because the climatic impacts of LULCC on economies are much

more complicated than those of greenhouse gases and anthropogenic aerosols.

Following the reviewer's suggestion, the effects of extreme temperature and temperature seasonal responses on economies are analyzed in the revision. The extreme temperature can be represented by day-to-day surface air temperature (SAT) variability, which indicates the magnitude of daily SAT fluctuations; greater SAT variability implies stronger heat and/or cold extremes. Kotz et al. (2021) recently found the macro-economic impacts of the day-to-day SAT variability, showing reduced economic growth in response to increased day-to-day SAT variability. Using the damage response function developed by Kotz et al. (2021), we quantified the economic impact of annual and seasonal day-to-day SAT variability changes induced by LULCC. The annual mean day-to-day SAT variability is measured as the intra-monthly standard deviation of daily SAT averaged across 12 months. The seasonal day-to-day SAT variability is averaged across three months of the season. The results show that extreme temperatures represented by day-to-day temperature variability are enhanced in poor economies in the tropics and subtropics (30°S–30°N) but reduced in wealthy economies at high latitudes in the Northern Hemisphere (north of 45°N), which deteriorates global economic inequality. Fig. R1 presents annual mean day-to-day SAT variability changes induced by combined biogeophysical (BGP) and biogeochemical (BGC) impacts of historical LULCC. Fig. R2 presents the associated economic impact. Fig. R3 decomposes the economic impact into contributions from four seasons, showing a major contribution from spring. There are two causes. First, the economic sensitivity to the day-to-day SAT variability change in spring is greater than that in the other three seasons (Kotz et al., 2021). Second, LULCC impacts on spring day-to-day SAT variability are prominent (Fig. R4).

References:

Kotz, M., Wenz, L., Stechemesser, A., Kalkuhl, M. & Levermann, A. Day-to-day temperature variability reduces economic growth. *Nat Clim Change* 1–7 (2021) doi:10.1038/s41558-020-00985-5.

Fig. R1. Spatial patterns of changes in annual mean day-to-day SAT variability during 1993–2012 due to combined BGP and BGC impacts of historical LULCC. Dots indicate where more than two-thirds of members agree on the sign of response.

Fig. R2. Country-level cumulative economic impacts via annual-mean day-to-day SAT variability changes from 1993 to 2012 due to combined BGP and BGC impacts of historical LULCC. e The ensemble median of the relative changes in GDP per capita (%) in 2012. **f** The corresponding probability level of the economic damage/benefit according to the IPCC uncertainty guidance. “Very likely”, “Likely”, and “More likely than not” indicate that more than 90%, two-thirds, and half of the members agree on the response, respectively. Countries and regions with missing values are shaded in grey.

Fig. R3. Country-level cumulative economic impacts via seasonal day-to-day SAT variability changes from 1993 to 2012 due to the combined BGP and BGC impacts of historical LULCC. a–d Relative changes in GDP per capita (%) in 2012 due to day-to-day SAT variability changes in boreal (a) spring, (b) summer, (c) autumn, and (d) winter. **e** Total economic impacts of the four seasons. **f** The economic impact of annual mean day-to-day SAT variability changes. Countries and regions with missing values are shaded in grey.

Fig. R4. Spatial patterns of seasonal day-to-day SAT variability changes (°C) during 1993–2012 due to the combined BGP and BGC impacts of historical LULCC. For boreal (a) spring, (b) summer, (c) autumn, and (d) winter. Dots indicate where more than two-thirds of members agree on the sign of response.

In the revision, Figs. R1&2 are included in Figs. 3&8. Figs. R3&4 are provided in Supplementary Figs. 5&6. The corresponding edits have been made in Lines 72–79 in the Introduction:

Previous studies using the temperature–growth response function only account for the economic impact of the change in annual mean temperature. Recent work revealed that day-to-day temperature variability has a greater economic influence than the annual mean temperature because dramatic economic losses occur on extremely hot and cold days^{25,30,31}. An extra degree Celsius of temperature variability reduces the growth rate of GDP per capita by 5% on average, featuring higher vulnerability in low-latitude and low-income countries²⁵. Historical LULCC has been found to contribute significantly to the observed greater day-to-day temperature variability as well^{13,16,32,33}, in addition to its impact on annual mean temperature^{9,12,18}.

Lines 162–171 and 236–266, showing the LULCC-induced changes in day-to-day SAT variability and the associated economic impacts:

Day-to-day SAT variability reflects the magnitude of daily temperature fluctuations. Greater SAT variability implies stronger daily heat or/and cold extremes⁴⁸. As shown in Fig. 3d–f, the BGC impact plays a leading role in LULCC-induced SAT variability changes. The SAT variability is decreased at high latitudes in the Northern Hemisphere (north of 60°N) but increased in the tropics and subtropics (30°S–30°N) (Fig. 3f). It has been documented that the warming due to increased atmospheric CO₂ concentrations leads to Arctic sea-ice loss and thus reduces the meridional temperature gradient in the extratropics due to Arctic amplification. This further weakens synoptic wave activities and results in decreased day-to-day SAT variability there^{49,50}. The increased SAT variability in the tropics is mainly caused by soil drying in response to the warming from the increased CO₂ concentrations^{49–51}.

Economic impacts of day-to-day SAT variability changes

The annual mean SAT changes capture a fraction of SAT impacts on economies. In addition to the annual mean SAT changes, the changes in day-to-day SAT variability have a greater economic influence²⁵. With the global maps of annual mean day-to-day SAT variability changes induced by BGC and BGP impacts of historical LULCC and their combination (Fig. 3), we investigate their corresponding influences on economies. Overall, the signs of the BGC and BGP impacts on economies are opposite in many countries (Fig. 8a and c). The combined impact on economies is regulated by the BGC impact due to its dominant role in affecting the day-to-day SAT changes (Figs. 3f and 8e). In comparison with countries in midlatitudes and high-latitudes, low-latitude countries with smaller seasonal temperature variations are damaged more given the increases in day-to-day SAT variability because they have more difficulty in adapting to greater temperature fluctuations²⁵ (Fig. 8e). In addition, low-latitude countries with poorer economies (Supplementary Fig. 4) are more vulnerable to the negative economic impact of the increased day-to-day SAT variability because of less insurance and risk-management practices²⁵. Therefore, many low-latitude and low-income countries (e.g., the majority of African and Southeast Asian countries) are greatly damaged from the increased day-to-day SAT variability imposed by the combined impact of LULCC (Fig. 8e). In contrast, the economies of many rich countries in the extratropics (e.g., Canada, the US, and Western European countries) benefit from reduced day-to-day SAT variability there. Quantitatively, the 80:20 and 90:10 ratios of the global economic gap are increased by +9.36% and +2.49%, respectively, due to the combined impact of LULCC. Therefore, in addition to the annual mean SAT impact, LULCC-induced day-to-day SAT variability changes further deteriorate global economic inequality.

We further decompose the impact of annual mean day-to-day SAT variability changes on the annual GDP per capita into economic contributions from day-to-day SAT variability changes in four seasons²⁵ (see Methods). It is shown that spring dominates the annual economic impacts in many countries (Supplementary Figs. 5 and 6). This is mainly due to greater economic sensitivities to spring day-to-day SAT variability changes compared with the other three seasons²⁵. In addition, LULCC impacts on spring day-to-day SAT variability are also prominent (Supplementary Fig. 5a). Overall, the total impact of seasonal day-to-day SAT variability changes on annual GDP per capita is comparable to that calculated by annual mean day-to-day SAT variability changes with slightly noticeable differences in the USA, Russia, and Finland (Supplementary Fig. 6e and f).

Lines 510–542 in the section of the Methods (i.e., estimates of economic impacts of day-to-day SAT variability changes):

Estimates of economic impacts of day-to-day SAT variability changes. In addition to annual mean SAT, we also calculate day-to-day SAT variability, which reflects the magnitude of daily temperature fluctuations; greater temperature variability implies stronger heat or/and cold extremes⁴⁸. The annual mean day-to-day SAT variability is calculated as the intra-monthly standard deviation of daily SAT averaged across 12 months of a year²⁵. The measure of day-to-day variability is seasonally adjusted because economic agents are already adapted to the seasonal cycle²⁵. Both heat and cold extremes are involved in day-to-day SAT variability. A recent study has documented a damage function describing country-level responses of the GDP per capita growth rate to every extra degree Celsius of the day-to-day SAT variability²⁵:

$$\Delta Growth_{LULCC} = \alpha \times \Delta TVAR_{LULCC} \quad (8)$$

where $\Delta TVAR_{LULCC}$ is the LULCC-induced country-level day-to-day SAT variability change and $\Delta Growth_{LULCC}$ is the associated change in the growth rate of GDP per capita. α is a negative parameter denoting the decrease in the growth rate of GDP per capita from an extra degree Celsius of day-to-day SAT variability (greater $|\alpha|$ implies greater damage). The parameter is dependent on the country-level seasonal temperature difference. The magnitude of α is greater in countries with smaller seasonal variations (e.g., tropical countries). Using the damage response function and then following the same process as the estimates of economic impacts of annual mean SAT changes, the economic impacts of day-to-day SAT variability changes can be derived with country-level day-to-day SAT variability changes induced by LULCC ($\Delta TVAR_{LULCC}$) calculated from CMIP6 simulations.

In addition to the annual mean seasonally-adjusted day-to-day SAT variability (averaged across 12 months), we also calculate seasonal day-to-day SAT variability averaged across three months of the season for boreal spring (March, April, May), summer (June, July, August), autumn (September, October, November), and winter (December, January, February). Then, the impacts of these seasonal day-to-day SAT variability changes induced by LULCC on annual GDP per capita are investigated according to a season-specific damage function developed by ref²⁵. These seasonal influences show their respective economic contributions to annual GDP per capita changes and can be summed to reflect annual cumulative economic impacts with the seasonal cycle. For a given day-to-day SAT variability, its economic impact varies in different seasons²⁵. Compared with other seasons, spring shows greater economic damage in response to increased day-to-day SAT variability²⁵. Economic growth in winter is insensitive to day-to-day SAT variability changes because economic agents may be sheltered from extreme weather effects by reducing outdoor activities²⁵.

The uncertainties of LUH2 land-use and ERA5 data have been discussed in Lines 285–290 and 438–444 in the revision:

There are some uncertainties in historical land-use forcing. To present the upper and lower bounds of the uncertainties in historical land-use forcing, LUH2 developed two extreme historical land-use reconstructions (“high” and “low” land-use) in terms of the cumulative wood harvested and total area of forests removed^{4,43}. The land-use uncertainties may affect the magnitude of LULCC-induced SAT changes. Nonetheless, the spatial patterns of the changes are kept unchanged^{9–18}.

Global gridded SAT in the factual world from 1993 to 2012 are derived from the European Centre for Medium-Range Weather Forecasts (ECMWF) Reanalysis version 5 (ERA5)³⁸ and the Modern-Era Retrospective analysis for Research and Applications Version 2 (MERRA2)⁴⁰. The former is used for the central estimate and the latter is used for the sensitivity test. They are interpolated to the $2.5^{\circ} \times 1.9^{\circ}$ resolution and then aggregated to the country level. The sensitivity test demonstrates the robust historical LULCC impacts despite the different SAT reanalysis data used (Figs. 5–7 and Supplementary Figs. 7–9).

Specific comments:

Line 18 “warming has been shown to harm economies in warm climates but provide benefits in cold climates.” Which are specific aspects in economies? And why?

Reply: We estimate the macro-economic impacts of temperature changes according to

the historical statistical relation of country-level annual GDP per capita growth rate and annual mean temperature (temperature–growth response function) (Burke et al., 2018). The response function empirically reflects that the growth rate of GDP per capita peaks at the optimal temperature and declines at higher or lower temperatures (Fig. R5). GDP per capita is the value of goods and services produced within a country in a year per person. The statistical relation is in accord with the empirical evidence that in the hottest and coldest areas on the earth, per capita economic productivity is low, and that, when other conditions are equal, economic productivity peaks at some intermediate temperature. This implies that temperature affects macro-economic growth by influencing agricultural yields, energy supply, labor productivity, and human health, as shown in previous studies (Carleton and Hsiang, 2016). Fig. R5 illustrates the bootstrapped temperature–growth response function. For warm countries with annual mean SAT warmer than the temperature optimum, warming moves country-level temperatures further from the temperature optimum, thus damaging economic growth there (e.g., Zambia, India, and Saudi Arabia). In contrast, the warming benefits economic growth for cool countries whose annual mean SAT is colder than the temperature optimum because it makes country-level temperatures closer to the temperature optimum (e.g., Iceland and the UK).

References:

- Burke, M., Davis, W. M. & Diffenbaugh, N. S. Large potential reduction in economic damages under UN mitigation targets. *Nature* **557**, 549–553 (2018).
 Carleton, T. A. & Hsiang, S. M. Social and economic impacts of climate. *Science* **353**, aad9837 (2016).

Fig. R5. Bootstrapped temperature–growth response function represented by 25th (left dashed curve), 50th (solid curve), and 75th (right dashed curve) percentiles of 1,000 members of the temperature optimum. Vertical lines overlaid on the curves are annual mean temperatures from factual observations with LULCC (brown) and the counterfactual world without the combined BGP and BGC impacts of LULCC (green) for some representative countries.

In the revision, Fig. R5 has been included as Fig. 4a. Owing to the word limit of the abstract (150 words or fewer), it is hard to present the details here but we clarify these in Lines 92–94, 175–186, and 457–461 instead:

The economic impact is measured by GDP per capita, showing the value of goods and

services produced within a country in a year per person.

Fig. 4 shows the combined impacts of LULCC on economies. For warm countries with annual mean SAT warmer than the temperature optimum (mostly between 30°S and 30°N), LULCC-induced warming makes country-level temperatures deviate further from the temperature optimum, thus damaging the economies there (e.g., Zambia, India, and Saudi Arabia). In contrast, the warming benefits the economic growth for cool countries whose annual mean SAT is colder than the temperature optimum (mostly north of 45°N) because it makes country-level temperatures closer to the temperature optimum (e.g., Iceland and the UK). For mid-latitude countries with annual mean SAT close to the temperature optimum (e.g., the USA and China), economic growth is insensitive to annual mean SAT changes there. These changes are regulated by the BGC impact of LULCC due to its dominant role in annual mean temperature changes (Supplementary Fig. 2a), except Canada, which is controlled by the BGP-induced cooling (Supplementary Fig. 3a).

The statistical relation is in accord with the empirical evidence that in the hottest and coldest areas on the earth, per capita economic productivity is low, and that, when other conditions are equal, economic productivity peaks at some intermediate temperature. This implies that temperature affects macro-economic growth by influencing agricultural yields, energy supply, labor productivity, and human health, as shown in previous studies²¹.

Line 21 “unlike anthropogenic aerosols, the climate effects of LULCC differ in latitudes and disturbed land surface types.” What is effect of anthropogenic aerosols?

Reply: Anthropogenic aerosols scatter incoming solar radiation and interact with clouds, masking some of warming induced by greenhouse gases. The spatial pattern of simulated temperature responses to anthropogenic aerosols shows global cooling (Zheng et al., 2020). In the revision, owing to the word limit of the abstract (150 words or fewer), it is hard to present the details here but we clarify this in Lines 47–51 in the introduction instead:

Owing to the spatial heterogeneity of the land surface, the BGP impact of LULCC on temperature spatially is not as homogeneous as warming induced by greenhouse gases and cooling resulting from aerosols⁹. It varies in latitudes and disturbed land surface types^{17–18}. For instance, the BGP impact of deforestation leads to warming in the tropics by reducing evapotranspiration but cooling in the extratropics because of increased surface albedo^{17–19}.

References:

Zheng, Y., Davis, S. J., Persad, G. G. & Caldeira, K. Climate effects of aerosols reduce economic inequality. *Nat Clim Change* **10**, 220–224 (2020).

Line 22 “we presented historical LULCC-induced temperature changes by multi-model climate simulations from the Coupled Model” please explain what is “LULCC-induced temperature” and which simulations?

Reply: The impact of historical LULCC on temperature is estimated by comparing two climate simulation experiments from CMIP6. They are the “historical” experiment

driven by historical natural and anthropogenic forcings and the “hist-noLu” experiment, which is identical to the former, except all land-use and land-cover are maintained at the preindustrial level. By comparing the “historical” with “hist-noLu” experiments, we isolate the impact of historical LULCC on temperature. The combined BGC and BGP impacts of historical LULCC on annual mean SAT is warming in most countries (Fig. R6).

Fig. R6. Spatial patterns of changes in annual mean SAT during 1993–2012 due to BGP and BGC impacts of historical LULCC. a–c The ensemble median of annual mean SAT changes (°C) induced by (a) BGP, (b) BGC, and (c) their combined impacts. Dots indicate where more than two-thirds of members agree on the sign of response.

In the revision, owing to the word limit of the abstract (150 words or fewer), it is hard to present the details of experiments here and thus we introduce them in Lines 102–109 instead:

We compute the changes in annual mean SAT and day-to-day SAT variability due to the BGP and BGC impacts of LULCC. The BGP impacts are estimated by directly comparing the modeled mean SAT and day-to-day SAT variability between two concentration-driven climate simulation experiments from the Coupled Model Intercomparison Project Phase 6 (CMIP6) (Supplementary Table 1). They are the “historical” experiment, which is driven by historical natural and anthropogenic forcings including historical LULCC from 1850 to 2014, and the “hist-noLu” experiment, which is identical to the former, except all land-use and land-cover are maintained at the 1850 level^{42,43}.

The combined biogeochemical and biogeophysical impacts of historical LULCC on annual mean SAT is warming in most countries, which has been stated in Lines 24–26 in the abstract in the revision:

Their combined effects on AMT result in warming in most countries, which harms poor economies in warm climates but benefits wealthy economies in cold climates.

Lines 25-26 “Historical LULCC resulted in strong cooling of up to...” please define a specific period in historical; “... ..some cooling/warming over tropics... ..” please make this clear.

Reply: The historical period of LULCC is from 1850 to 2014, according to the “historical” and “hist-noLu” experiments used to estimate LULCC impacts. Both of them are conducted from 1850 to 2014. The “historical” experiment is driven by historical natural and anthropogenic forcings, including historical LULCC from 1850 to 2014. However, all land-use and land-cover are maintained at the 1850 level in the “hist-noLu” experiment during the simulation from 1850 to 2014. We specified the historical period in Lines 22–24 in the abstract in the revision:

In this work, based on multi-model simulations from the Coupled Model Intercomparison Project Phase 6, contrasting influences of biogeochemical and biogeophysical impacts of historical (1850–2014) LULCC on economies are found.

In the original manuscript, the BGP impact of LULCC results in strong cooling over North America and Eurasia while warming over the Amazon and central Africa (Fig. R6a). In the revision, owing to the word limit of the abstract (150 words or fewer), it is hard to present the details here but we clarify these in Lines 152–154 instead:

Consistent with previous studies^{17–19}, the BGP impact of deforestation leads to significant cooling in North America and Eurasia mainly because of increased surface albedo but warming over the Amazon and central Africa driven by reduced evapotranspiration (Fig. 3a).

In the Introduction, please add some previous studies about LULCC-induced temperature changes and say why “LULCC-induced temperature” is important.

Reply: We have added some previous studies and provided more explanations of the importance of the LULCC-induced temperature changes, as indicated in the response to the major comments.

Lines 253-256 “With all forcing agents can be attributed to historical LULCC.” This sentence should be mentioned in the introduction.

Reply: Thanks for the suggestion. It has been mentioned in Lines 54–55 in the introduction:

When BGP and BGC processes are considered together, up to 40% ($\pm 16\%$) of the present-day global anthropogenic warming can be attributed to historical LULCC²⁰.

Line 103 In fig1, is there no urban land fraction showing? The period is from 1850-

2014 in figs 1-2, which should be changed into the period of 1991-2010 for economy responses as in fig 3?

Reply: The fraction of urban land is much smaller than that of other land cover types. We have made modifications to Fig.1a to see the fraction of urban land more clearly (Fig. R7).

Fig. R7. Global historical LULCC from 1850 to 2014. Land-cover fractions (relative to global land except for Antarctica) of primary vegetation (dark green), secondary vegetation (light green), cropland (orange), grazing land (yellow), and urban land (pink). The fractions of the five land-cover types add up to less than 1 due to the coverage of ice and water (shown in the white area).

The “historical” and “hist-noLu” experiments in CMIP6 are both conducted from 1850 to 2014. The “historical” experiment is driven by historical natural and anthropogenic forcings, including historical LULCC from 1850 to 2014. However, all land-use and land-cover are maintained at the 1850 level in the “hist-noLu” experiment during the simulation from 1850 to 2014. Therefore, land-use and land-cover differences between the two experiments during the study period of 1991–2010 actually show historical LULCC since 1850. Therefore, there is a need to show the evolution of historical LULCC since 1850 in Figs. 1–2. To reduce interannual variability as much as possible, we compute 5-year running averages during 1850–2014, yielding annual-mean temperatures from 1852–2012. The temperature in 2012 refers to the annual mean temperature during 2010–2014. The reason that we focus on LULCC-induced changes in SAT and associated economic impacts in the last two decades from 1993 to 2012 is that the socioeconomic data of most countries are available for the recent decades only. Note that the study period is changed to the 1993–2012 in the revision because we have changed the running-average strategy from 9-year to 5-year to avoid over-smoothing for our 20-year study period. The results do not change with this adjustment. We have made edits in Lines 103–109 and 406–413 to avoid confusion:

The BGP impacts are estimated by directly comparing the modeled mean SAT and day-to-day SAT variability between two concentration-driven climate simulation experiments from the Coupled Model Intercomparison Project Phase 6 (CMIP6) (Supplementary Table 1). They are the “historical” experiment, which is driven by

historical natural and anthropogenic forcings including historical LULCC from 1850 to 2014, and the “hist-noLu” experiment, which is identical to the former, except all land-use and land-cover are maintained at the 1850 level^{42,43}.

To reduce the interannual variability as much as possible, we calculate the 5-year running averages of SAT. For instance, SAT in 2012 refers to the annual mean SAT during 2010–2014. We focus on the impacts of historical LULCC on SAT and economies during the study period from 1993 to 2012, given the socioeconomic data of most countries dating back to just recent decades³⁹. Note that land-use and land-cover differences between the “historical” and “hist-noLu” experiments during the study period actually show historical LULCC since 1850 because all land-use and land-cover are maintained at the 1850 level in the “hist-noLu” experiment during the simulation from 1850 to 2014.

Lines 130-131 “Greenland and central Africa experienced some warming induced by LULCC.” What LULCC in Greenland and central Africa? How this LULCC induced the warming with bio-geophysical-chemical processes?

Reply: From 1850 to 2014, central Africa exhibited increased fractional coverage of cropland and grazing land with significant decreases in primary vegetation (Fig. R8). The BGP impact of the deforestation reduces evapotranspiration, leading to warming there (Fig. R6a), consistent with previous studies (e.g., Perugini et al., 2017). Minor LULCC occurred in Greenland, which implies that the warming there results from LULCC-induced changes in large-scale atmospheric circulation. Due to significant cooling over the mid-latitudes of the Northern Hemisphere (Fig. R6a), the upper-troposphere westerly jet is weakened north because of the decreased meridional thermal gradient there, following the principle of thermal wind adjustment (Huang et al., 2020). The westerly jet serves as an obstacle to the north–south airflow exchange. Thus, the weakened westerly jet induced by LULCC promotes warm advection in Greenland, leading to warming there (Fig. R6a). This feature indicates that LULCC results in the negative phase of the North Atlantic Oscillation (NAO), which has been regarded as a key driver of climate change in Greenland (Hanna et al., 2012). As shown in Fig. R6b, the BGC impact of LULCC leads to warming globally due to the emission of CO₂, a long-lived greenhouse gas well mixed in the atmosphere.

References:

- Perugini, L. et al. Biophysical effects on temperature and precipitation due to land cover change. *Environ Res Lett* **12**, 053002 (2017).
- Huang, H. et al. Assessing Global and Regional Effects of Reconstructed Land-Use and Land-Cover Change on Climate since 1950 Using a Coupled Land–Atmosphere–Ocean Model. *J Climate* **33**, 8997–9013 (2020).
- Hanna, E., Mernild, S. H., Cappelen, J. & Steffen, K. Recent warming in Greenland in a long-term instrumental (1881–2012) climatic context: I. Evaluation of surface air temperature records. *Environ Res Lett* **7**, 045404 (2012).

Fig. R8. Spatial patterns of historical LULCC from 1850 to 2014 (% , 2014 minus 1850). a Primary vegetation. b Secondary vegetation. c Cropland. d Grazing land. e Urban land.

In the revision, Figs. R6&R8 has been provided in Figs. 2&3. More explanations have been added Lines 127–132 and 152–159:

Humans have disturbed a majority of primary vegetation globally (Fig. 2). Secondary vegetation showed significant increases over mid-latitude Eurasia and the eastern USA. Cropland expansion was mainly distributed around the Great Lakes of North America, south of the Amazon, central Africa, north of the Caspian Sea, and India. The expansion of grazing land was more significant, mainly located in the USA, south of the Amazon, central and southern Africa, central Asia, and Australia.

Consistent with previous studies^{17–19}, the BGP impact of deforestation leads to significant cooling in North America and Eurasia mainly because of increased surface albedo but warming over the Amazon and central Africa driven by reduced evapotranspiration (Fig. 3a). The warming over Greenland where minor LULCC exists is related to the atmospheric circulation changes induced by non-local LULCC. Strong cooling at midlatitudes in the Northern Hemisphere weakens the upper-troposphere westerly jet and thus results in warm advection in Greenland⁴⁷. The BGC impact of LULCC leads to warming globally due to the emission of CO₂, a long-lived greenhouse gas well mixed in the atmosphere (Fig. 3b).

Lines 143-145 “we found that countries of low latitude were above the temperature optimum, while those of high latitude were below the optimum.” Clarify low/high latitude.

Reply: Warm countries with annual mean SAT warmer than the temperature optimum are mostly in low latitudes between 30°S and 30°N. Cool countries whose annual mean SAT is colder than the temperature optimum are mostly in high latitudes north of 45°N. We have clarified these in Lines 175–182 in the revision:

Fig. 4 shows the combined impacts of LULCC on economies. For warm countries with annual mean SAT warmer than the temperature optimum (mostly between 30°S and 30°N), LULCC-induced warming makes country-level temperatures deviate further from the temperature optimum, thus damaging the economies there (e.g., Zambia, India, and Saudi Arabia). In contrast, the warming benefits the economic growth for cool

countries whose annual mean SAT is colder than the temperature optimum (mostly north of 45°N) because it makes country-level temperatures closer to the temperature optimum (e.g., Iceland and the UK).

In the results section: the effects of the extreme temperature and seasonal cycles on the economy were considered in different countries? If this included, will change the conclusions you found now?

Reply: Thanks for the constructive suggestion. The effects of temperature extremes and seasonal cycles on economies were not considered in the original manuscript because the temperature–growth response function used was built on the historical statistical relation of country-level annual GDP per capita growth rate and annual mean temperature (Burke et al., 2018). In the revised manuscript, in addition to the annual-mean temperature, we explore the impacts of temperature extremes and their seasonal cycles on economies using a damage response function recently developed by Kotz et al. (2021). The results show that extreme temperatures represented by day-to-day temperature variability are enhanced in poor economies but reduced in wealthy economies, which deteriorates global economic inequality. Decomposing the economic impacts into contributions from four seasons, major contributions from spring are shown. Since the BGC impact of LULCC has been included in the revised manuscript in addition to the BGP impact, the combined (BGP+BGC) impact of LULCC on annual mean temperature also increases global economic inequality. Therefore, in addition to the economic impact from annual mean SAT changes, day-to-day variability changes further enlarge global economic inequality. We have made corresponding edits in the revision, as indicated in the response to the major comment.

References:

- Burke, M., Davis, W. M. & Diffenbaugh, N. S. Large potential reduction in economic damages under UN mitigation targets. *Nature* **557**, 549–553 (2018).
- Kotz, M., Wenz, L., Stechemesser, A., Kalkuhl, M. & Levermann, A. Day-to-day temperature variability reduces economic growth. *Nat Clim Change* 1–7 (2021) doi:10.1038/s41558-020-00985-5.

Line 152, why were India and Canada selected?

Reply: To show the opposite BGP impacts of LULCC on economies in warm poor countries and cool rich countries, we select two representative countries (featuring large but contrasting economic impacts) according to different background climates and national economic strengths. Canada is a developed country with a cool climate while India is a developing country with a warm climate. Despite different national economic strengths, India and Canada were among the top 10 countries in GDP in 2020. In the revision, we estimate both the BGP and BGC impacts of LULCC. In India, the BGC-induced warming exceeds the BGP-induced cooling impact, thus moving the country-level annual-mean temperature further from the temperature optimum, leading to economic damages. In Canada, the BGP-induced cooling exceeds the BGC-induced warming, thus moving the country-level annual-mean temperature further from the temperature optimum and resulting in economic damages too. We intend to show different economic responses to LULCC, so the UK, which shows economic benefits from the combined (BGP+BGC) impacts of LULCC, is selected to replace Canada. The UK is also a developed country with a cool climate, among the top 10 countries in GDP

in 2020. We have made edits in Lines 187–196 in the revision:

The opposite economic impacts of LULCC-induced warming on warm and cool countries may aggravate global economic inequality (Fig. 4) because most poor countries are in the low latitudes of the warm climate, but rich countries generally situate in temperate and cool climates^{27–29} (Supplementary Fig. 4). For example, for India (Fig. 4c and d), a developing country with a warm climate, the net warming impact of LULCC decreases the growth rate of GDP per capita annually, leading to cumulative economic damage of up to –6.35% (–0.69 to –12.20% in the 25th–75th range) in 2012. In comparison, the UK, a developed country with a cool climate, shows annual increases in the growth rate of GDP per capita because of warming, resulting in cumulative economic benefits of up to +1.60% (+0.34 to +3.91% in the 25th–75th range) in 2012 (Fig. 4e and f).

Lines 175-178 “For example, owing to the annual small warming impacts from the LULCC, GDP per capita of Greenland and Iceland in cool climates were estimated to gain in 2010, while that of warm countries over the southwest of Sahara were estimated to lose.” Why does this happen?

Reply: Greenland and Iceland are in cool climates cooler than the temperature optimum, so additional warming moves their country-level annual-mean temperatures closer to the optimum, thus benefitting economic growth there. In contrast, countries over the southwest of Sahara are in warm climates warmer than the temperature optimum, so additional warming leads to country-level temperatures further from the optimum, thus damaging economic growth there. Fig. 4a has been added to illustrate the bootstrapped temperature–growth response function which reflects the response of economic growth to annual-mean temperature (Fig. R5). Corresponding edits have been made in Lines 175–186 in the revision:

Fig. 4 shows the combined impacts of LULCC on economies. For warm countries with annual mean SAT warmer than the temperature optimum (mostly between 30°S and 30°N), LULCC-induced warming makes country-level temperatures deviate further from the temperature optimum, thus damaging the economies there (e.g., Zambia, India, and Saudi Arabia). In contrast, the warming benefits the economic growth for cool countries whose annual mean SAT is colder than the temperature optimum (mostly north of 45°N) because it makes country-level temperatures closer to the temperature optimum (e.g., Iceland and the UK). For mid-latitude countries with annual mean SAT close to the temperature optimum (e.g., the USA and China), economic growth is insensitive to annual mean SAT changes there. These changes are regulated by the BGC impact of LULCC due to its dominant role in annual mean temperature changes (Supplementary Fig. 2a), except Canada, which is controlled by the BGP-induced cooling (Supplementary Fig. 3a).

Line 179 how define the poor (rich) country?

Reply: According to the World Bank economic data, global GDP per capita in 2012 was approximately 10000 US\$₂₀₁₀, which is used to distinguish poor and rich countries. Countries with GDP per capita greater than 10000 US\$₂₀₁₀ are grouped into wealthier countries. Otherwise, they are grouped into poorer countries. Fig. R9 shows the spatial location of wealthier and poorer countries.

Fig. R9. Spatial patterns of country-level GDP per capita. Blue/red color is split at the global GDP per capita (approximately 10000 US\$₂₀₁₀). Countries and regions not analyzed in our study are shaded in grey. Countries with GDP per capita greater than 10000 US\$₂₀₁₀ are grouped into wealthier countries (in red). Otherwise, they are grouped into poorer countries (in blue).

In the revision, Fig. R9 has been included in Supplementary Fig. 4b. We have defined poor and rich countries in Lines 805–808 in the revision:

Blue/red color is split at the global GDP per capita in 2012 (approximately 10000 US\$₂₀₁₀)³⁹. Countries with GDP per capita greater than 10000 US\$₂₀₁₀ are grouped into wealthier countries (in red). Otherwise, they are grouped into poorer countries (in blue).

In the uncertainties section, some uncertainties of the land-use and land-cover data and ERA5 should be included.

Reply: Uncertainties of the LUH2 land-use data and ERA5 reanalysis data have been considered in the revision, as indicated in the response to the major comments. Figs. 5–7 and Supplementary Figs. 7–9 have been included to present the consistent economic impacts of LULCC constrained by different SAT reanalysis data (ERA5 and MERRA2).

Line 296 “the high-resolution requirement” how was the high-resolution defined?

Reply: In this study, all CMIP6 simulations are interpolated to the same horizontal resolution of 2.5°×1.9° (longitude×latitude), which is the coarsest grid in the selected global climate models. In the revision, “the high-resolution requirement” has been removed.

Reply to the comments by Reviewer #2

We thank the reviewer for the comments and suggestions to improve our manuscript. Below are our point-by-point responses to these comments. The reviewer's comments are in italics, our responses are in normal font, and manuscript revisions are in blue.

In their paper “Global Economic Inequality Reduced by Historical Land-Use and Land-Cover Change”, the authors use a relationship between annual mean temperature and economic success to attribute changes in countries’ GDP to historical land use – or, more specifically, to the strictly biophysical impact of historical land use on annual mean temperatures. They use two sets of CMIP6 model experiments – a “regular” historical simulation and a simulation that does not have historical land use (but has prescribed CO₂ concentrations, thus only captures the biophysical impacts of historical land use), as well as observations of historical climate, to determine how historical land use has impacted temperatures. They then apply their temperature-economics relationship to this temperature change to convert the temperature change to an economic change. They find that biophysical effect of historical land use has reduced global economic inequality, as the biophysical effects of historical land use have generally been to cool, which would benefit lower income countries in warm regions, and be detrimental to countries in cooler regions (more of which are higher income).

While I have not methodological problem with the study, I do have rather fundamental concerns about the conclusions the authors draw which may preclude the work from publication in this journal. The authors are aware of these issues – they mention most of them in their discussion at the end of the manuscript – but do not demonstrate in any way that these factors don’t change their conclusions. It is my feeling that these major limitations to their approach need to be addressed systematically in some way prior to publication, or the authors need to greatly reduce the strength of their conclusions.

Specifically, my concerns are that:

*- Land use and land cover change (LULCC) have historically had *as large* biogeochemical effects (i.e., impacts on the carbon cycle) as biogeophysical effects (the latter of which are explored here), globally averaged. However, these signals historically have been of opposite sign, largely cancelling out in the global mean. The authors conclude that LULCC has reduced global economic inequality, but they need to quantify or at least estimate how much the biogeochemical effect of LULCC would have changed economic inequality. Have the biogeochemical effects of LULCC increased economic inequality? Moreover, the biogeophysical and biogeochemical effects don’t operate on the same spatial scale; the carbon cycle impacts of LULCC are global, while the biogeophysical effects are mostly local/regional, so since economic inequality is also spatially heterogenous, it is possible the authors’ conclusion still holds, but it needs to be demonstrated!*

Reply: Thanks for the valuable comments. In the revision, in addition to the biogeophysical (BGP) impacts, the biogeochemical (BGC) impacts of LULCC on annual mean temperature and associated economic impacts have been included. Their combination (BGC+BGP) is also presented. Owing to the same atmospheric CO₂ concentrations prescribed in the “historical” and “hist-noLu” experiments, the modeled surface air temperature (SAT) difference between the two experiments only figures out the BGP impact of LULCC. Note that there is not a set of CMIP6 experiments that

allows us to directly evaluate the BGC impacts of historical LULCC. Here, to estimate the BGC impacts, the following steps are performed. First, the accumulated increase in atmospheric CO₂ concentrations induced by LULCC since 1850 is computed by comparing the global net land-atmosphere CO₂ fluxes between the “historical” and “hist-noLu” experiments and using a CO₂ pulse response function (Ward et al., 2012, 2014). The results show that the historical LULCC contribution to atmospheric CO₂ concentrations is approximately 25.3 ppm (the median of ensemble members) from 1850 to 2014 (Fig. R1), an increase of 9% relative to the atmospheric CO₂ concentration in 1850. Second, the CMIP6 “1pctCO2” experiment, where CO₂ concentrations are increased gradually at a rate of 1% per year since 1850, is used to isolate the SAT response to the increased CO₂ concentrations since 1850. For example, given an increase of 9% in 2014 relative to 1850, the corresponding LULCC-induced SAT differences of the BGC impact in 2014 are calculated as the SAT in the 9th year after 1850 minus that in 1850 in the “1pctCO2” experiment. This method accounts for different climate sensitivities in each model to the arising CO₂ concentrations and maps the global distribution of corresponding SAT changes. The derived atmospheric CO₂ emissions and associated SAT changes induced by LULCC are comparable to those documented in previous studies (Ward et al., 2014; Windisch et al., 2021). We further estimate the associated economic impact of the BGC effect of LULCC on SAT. Updated results show that although LULCC results in BGP cooling, it also leads to BGC warming, which dominates SAT changes over most regions of the world (Fig. R2). Thus, there are contrasting influences of the BGP and BGC impacts on economies (Fig. R3). The BGC-dominated warming harms poor countries with warm climates but benefits wealthy countries with cold climates, so global economic inequality is increased when combining the two impacts. This study suggests that the climate feedback of LULCC leads to potential declines in global economic equitable and sustainable development.

References:

- Ward, D. S., Mahowald, N. M. & Kloster, S. Potential climate forcing of land use and land cover change. *Atmos Chem Phys* **14**, 12701–12724 (2014).
- Ward, D. S. et al. The changing radiative forcing of fires: global model estimates for past, present and future. *Atmos Chem Phys* **12**, 10857–10886 (2012).
- Windisch, M. G., Davin, E. L. & Seneviratne, S. I. Prioritizing forestation based on biogeochemical and local biogeophysical impacts. *Nat Clim Change* **11**, 867–871 (2021).

Fig. R1. Cumulative increases in atmospheric CO₂ concentrations due to historical

LULCC from 1850 to 2014. The black line and corresponding brown shading denote the median and 25th–75th percentile range of ensemble members.

Fig. R2. Spatial patterns of changes in annual mean SAT during 1993–2012 due to BGP and BGC impacts of historical LULCC. a–c The ensemble median of annual mean SAT changes (°C) induced by (a) BGP, (b) BGC, and (c) their combined impacts. Dots indicate where more than two-thirds of members agree on the sign of response.

Fig. R3. Country-level cumulative economic impacts via annual-mean SAT changes from 1993 to 2012 due to BGP and BGC impacts of historical LULCC. a, c, e The ensemble median of the relative changes in GDP per capita (%) in 2012 induced by **(a)** BGP, **(c)** BGC, and **(e)** their combined impacts. **b, d, f** The corresponding probability level of the economic damage/benefit according to the IPCC uncertainty guidance⁵² for **(b)** BGP, **(d)** BGC, and **(f)** their combined impacts. “Very likely”, “Likely”, and “More likely than not” indicate that more than 90%, two-thirds, and half of the members agree on the response, respectively. Countries and regions with missing values are shaded in grey.

In the revision, Fig. R1–R3 are provided in Figs. 1, 3, and 5. Corresponding edits have been made in Lines 80–86 and 102–114 in the introduction:

It has been documented that long-term economic sustainable development is limited due to substantial disruption to the ecosystem from historical LULCC despite short-term regional economic growth with the gains of agricultural/industrial commodities^{34–37}. Nonetheless, the economic impact of historical LULCC through climate feedback is still unclear. This is because the climatic impacts of LULCC on economies are much more complicated than those of greenhouse gases and anthropogenic aerosols. LULCC has both BGP and BGC impacts on climate, and the BGP impacts vary in latitudes and land surface types.

We compute the changes in annual mean SAT and day-to-day SAT variability due to the BGP and BGC impacts of LULCC. The BGP impacts are estimated by directly comparing the modeled mean SAT and day-to-day SAT variability between two concentration-driven climate simulation experiments from the Coupled Model Intercomparison Project Phase 6 (CMIP6) (Supplementary Table 1). They are the “historical” experiment, which is driven by historical natural and anthropogenic forcings including historical LULCC from 1850 to 2014, and the “hist-noLu” experiment, which is identical to the former, except all land-use and land-cover are maintained at the 1850 level^{42,43}. The corresponding BGC impacts are estimated according to the accumulated increase in atmospheric CO₂ concentrations induced by LULCC since 1850 by comparing the global net land-atmosphere CO₂ flux between the “historical” and “hist-noLu” experiment²⁰. Then, the CMIP6 “1pctCO2” experiment is used to isolate the SAT response of the increased CO₂ concentration since 1850 (see Methods). The BGP and BGC impacts can be added up to obtain their combined impacts^{20,44,45}.

Lines 152–161 explaining the BGP and BGC impacts of LULCC on annual mean SAT:

Consistent with previous studies^{17–19}, the BGP impact of deforestation leads to significant cooling in North America and Eurasia mainly because of increased surface albedo but warming over the Amazon and central Africa driven by reduced evapotranspiration (Fig. 3a). The warming over Greenland where minor LULCC exists is related to the atmospheric circulation changes induced by non-local LULCC. Strong cooling at midlatitudes in the Northern Hemisphere weakens the upper-troposphere westerly jet and thus results in warm advection in Greenland⁴⁷. The BGC impact of LULCC leads to warming globally due to the emission of CO₂, a long-lived greenhouse gas well mixed in the atmosphere (Fig. 3b). Over most continents of the world, BGC-

induced warming dominates the combined SAT changes, except the central and eastern parts of North America, Central Asia, and East Europe (Fig. 3c)^{20,44,45}.

Lines 202–213 showing associated economic impacts of the BGP and BGC impacts of LULCC on annual mean SAT:

Fig. 5 shows the global country-level cumulative economic impacts of the annual mean SAT changes from 1993 to 2012 caused by the respective BGP and BGC impacts and their combination of historical LULCC. Contrasting economic influences of individual BGP and BGC impacts of LULCC are found owing to their opposite influences on annual mean SAT (Fig. 3). Most low-latitude countries with warm climates (e.g., the majority of African and Southeast Asian countries) are experiencing positive economic impacts from the BGP cooling, but negative economic impacts from the BGC warming (Fig. 5a and c). Many cool-climate countries in the high latitudes of the Northern Hemisphere (e.g., Russia, Canada, and Norway) experience damage from the BGP cooling but benefit from the BGC warming. The combined impact of LULCC on economies is generally controlled by the BGC warming (Fig. 5e), except for some countries with cool climates over northern mid-latitudes cooled by the BGP effect (Fig. 3c), which decreases their economic growth (e.g., Canada, Sweden, and Finland).

Lines 352–361 and 376–405 introducing the methodology to estimate the BGP and BGC impacts of LULCC on SAT:

The modeled SAT differences between the two experiments only isolate the BGP impact of LULCC because of the same CO₂ concentrations prescribed in the two concentration-driven experiments without climate feedback of LULCC-induced CO₂ emissions (i.e., the BGC impact)⁴³. Despite the same atmospheric CO₂ concentrations, carbon cycle processes (e.g., the land-atmosphere CO₂ flux) can be simulated according to the land surface and climate conditions in the two experiments⁴³. To derive the BGC effect of historical LULCC, the CMIP6 “1pctCO₂” experiment is used. The “1pctCO₂” experiment, where CO₂ concentration is increased gradually at a rate of 1% per year since 1850, is used to isolate the SAT response of accumulated LULCC-induced increases in CO₂ concentrations since 1850 (see estimates of LULCC impacts on SAT below)^{42,61,62}.

Estimates of LULCC impacts on SAT. The BGP impact of historical LULCC on SAT is estimated by directly comparing modeled SAT between the “historical” and “hist-noLu” experiments. The BGC impact of LULCC on SAT is estimated by calculating accumulated LULCC-induced increases in CO₂ concentrations since 1850²⁰ and applying them to the “1pctCO₂” experiments. Although the “historical” and “hist-noLu” experiments in CMIP6, unlike emission-driven simulations, are concentration-driven simulations with atmospheric CO₂ concentrations prescribed, some GCMs still compute the land-atmosphere CO₂ fluxes offline accounting for atmospheric feedback⁴³. As such, global yearly LULCC-induced CO₂ emissions are calculated with the annual differences in the global net land-atmosphere CO₂ fluxes between the “historical” and “hist-noLu” experiments^{20,43,63}. It accounts for the net exchange of CO₂ between the land and atmosphere associated with LULCC (e.g., deforestation and reforestation)^{20,64}. It also includes the BGP feedback of LULCC because of the climatic differences between the two experiments due to the BGP impact^{20,43,64,65}. CO₂ is chemically inert in the atmosphere, but over time, the airborne fraction of emitted CO₂ concentration

decreases with carbon uptake of ocean and land, the highest airborne fraction for the most recent CO₂ emissions. Following prior studies^{20,64}, we calculate the airborne fraction of the yearly LULCC-induced CO₂ emissions since 1850 using a CO₂ pulse response function that presents the residual fraction of CO₂ over time due to carbon uptake⁶⁴. Annual LULCC-induced CO₂ emissions are multiplied by corresponding airborne fractions and then summed over time since 1850 to yield accumulated atmospheric CO₂ concentrations at present. Here, the historical LULCC contribution to atmospheric CO₂ concentration is approximately 25.3 ppm (the median of ensemble members) from 1850 to 2014, an increase of 9% relative to the atmospheric CO₂ concentration in 1850. The SAT responses to the accumulated CO₂ concentrations are estimated by the “1pctCO2” experiment, which accounts for different climate sensitivities in each model to the increasing CO₂ concentrations and maps the global distribution of corresponding SAT changes^{61,62}. For example, given an increase of 9% in 2014 relative to 1850, the corresponding LULCC-induced SAT differences of the BGC impact in 2014 are calculated as the SAT in the 9th year after 1850 minus that in 1850 in the “1pctCO2” experiment. The BGP and BGC impacts can be arithmetically summed to obtain their combined impacts^{20,44,45,66}. Our derived results of LULCC-induced CO₂ emissions and SAT changes are comparable to those of previous studies^{17–20,45,47,49,54,66}.

- LULCC has a much stronger impact on temperatures and the water/energy cycles on seasonal timescales, rather than annual mean timescales. In addition, LULCC would likely have had a large impact on changes in extremes, and changes in extremes would, I would expect anyhow, have had greater economic impact than changes in annual mean temperatures. However, the authors' approach of using annual mean temperature to determine economic benefit would completely miss this, and I would expect this to be the larger part of the signal. How can the authors be confident their historical annual mean T / annual mean GDP relationship can be applied to LULCC induced perturbations of annual mean T?

Reply: The temperature–growth response function used to estimate the economic impact of annual mean temperature changes was built on the historical statistical relation of country-level annual GDP per capita growth rate and annual mean temperature (Burke et al., 2015, 2018). GDP per capita is the value of goods and services produced within a country in a year per person. They found that per capita economic productivity peaks at the optimal annual mean temperature and declines at higher or lower temperatures (statistically significant). The statistical relation is in accord with the empirical evidence that in the hottest and coldest areas on the earth, per capita economic productivity is low, and that, when other conditions are equal, economic productivity peaks at some intermediate temperature. This implies that temperature affects macro-economic growth by influencing agricultural yields, energy supply, labor productivity, and human health, as shown in previous studies (Carleton and Hsiang, 2016). This empirical relationship has been widely applied to estimate the economic impacts of annual mean temperature perturbations from greenhouse gases and aerosols (Diffenbaugh and Burke, 2019; Zheng et al., 2020). In this study, we further use it to quantify LULCC induced perturbations.

As the reviewer said, this function does ignore the effects of extreme temperature and seasonal cycles on the economy. Following the reviewer's suggestion, in addition to the annual mean effect, the effects of extreme temperature and its seasonal variation on the economy are considered in the revision. The extreme temperature can be represented

by day-to-day SAT variability, which indicates the magnitude of daily SAT fluctuations; greater SAT variability implies stronger heat or/and cold extremes. Kotz et al. (2021) recently found the macro-economic impacts of the day-to-day SAT variability, showing reduced economic growth in response to increased day-to-day SAT variability. Using the damage response function developed by Kotz et al. (2021), we quantified the economic impact of annual and seasonal day-to-day SAT variability changes induced by LULCC. The annual mean day-to-day SAT variability is measured as the intra-monthly standard deviation of daily SAT averaged across 12 months. The seasonal day-to-day SAT variability is averaged across three months of the season. The results show that extreme temperatures represented by day-to-day temperature variability are enhanced in poor economies in the tropics and subtropics (30°S–30°N) but reduced in wealthy economies at high latitudes in the Northern Hemisphere (north of 45°N), which deteriorates global economic inequality. Fig. R4 presents annual mean day-to-day SAT variability changes induced by combined BGP and BGC impacts of historical LULCC. Fig. R5 presents the associated economic impact. Fig. R6 decomposes the economic impact into contributions from four seasons, showing a major contribution from spring. There are two causes. First, the economic sensitivity to the day-to-day SAT variability change in spring is greater than that in the other three seasons (Kotz et al., 2021). Second, LULCC impacts on spring day-to-day SAT variability are prominent (Fig. R7).

References:

- Burke, M., Hsiang, S. M. & Miguel, E. Global non-linear effect of temperature on economic production. *Nature* **527**, 235–239 (2015).
- Burke, M., Davis, W. M. & Diffenbaugh, N. S. Large potential reduction in economic damages under UN mitigation targets. *Nature* **557**, 549–553 (2018).
- Carleton, T. A. & Hsiang, S. M. Social and economic impacts of climate. *Science* **353**, aad9837 (2016).
- Diffenbaugh, N. S. & Burke, M. Global warming has increased global economic inequality. *Proc National Acad Sci* **116**, 201816020 (2019).
- Zheng, Y., Davis, S. J., Persad, G. G. & Caldeira, K. Climate effects of aerosols reduce economic inequality. *Nat Clim Change* **10**, 220–224 (2020).
- Kotz, M., Wenz, L., Stechemesser, A., Kalkuhl, M. & Levermann, A. Day-to-day temperature variability reduces economic growth. *Nat Clim Change* 1–7 (2021) doi:10.1038/s41558-020-00985-5.

Fig. R4. Spatial patterns of changes in annual mean day-to-day SAT variability during 1993–2012 due to combined BGP and BGC impacts of historical LULCC. Dots indicate where more than two-thirds of members agree on the sign of response.

Fig. R5. Country-level cumulative economic impacts via annual-mean day-to-day SAT variability changes from 1993 to 2012 due to combined BGP and BGC impacts of historical LULCC. e The ensemble median of the relative changes in GDP per capita (%) in 2012. **f** The corresponding probability level of the economic damage/benefit according to the IPCC uncertainty guidance. “Very likely”, “Likely”, and “More likely than not” indicate that more than 90%, two-thirds, and half of the members agree on the response, respectively. Countries and regions with missing values are shaded in grey.

Fig. R6. Country-level cumulative economic impacts via seasonal day-to-day SAT variability changes from 1993 to 2012 due to the combined BGP and BGC impacts of historical LULCC. a–d Relative changes in GDP per capita (%) in 2012 due to day-to-day SAT variability changes in boreal (a) spring, (b) summer, (c) autumn, and (d) winter. **e** Total economic impacts of the four seasons. **f** The economic impact of annual mean day-to-day SAT variability changes. Countries and regions with missing values are shaded in grey.

Fig. R7. Spatial patterns of seasonal day-to-day SAT variability changes (°C) during 1993–2012 due to the combined BGP and BGC impacts of historical LULCC. For boreal (a) spring, (b) summer, (c) autumn, and (d) winter. Dots indicate where more than two-thirds of members agree on the sign of response.

In the revision, Figs. R4&R5 are included in Figs. 3&8. Figs. R6&R7 are provided in Supplementary Figs. 5&6. The corresponding edits have been made in Lines 72–79 in the introduction:

Previous studies using the temperature–growth response function only account for the economic impact of the change in annual mean temperature. Recent work revealed that day-to-day temperature variability has a greater economic influence than the annual mean temperature because dramatic economic losses occur on extremely hot and cold days^{25,30,31}. An extra degree Celsius of temperature variability reduces the growth rate of GDP per capita by 5% on average, featuring higher vulnerability in low-latitude and low-income countries²⁵. Historical LULCC has been found to contribute significantly to the observed greater day-to-day temperature variability as well^{13,16,32,33}, in addition to its impact on annual mean temperature^{9,12,18}.

Lines 162–171 and 236–266, showing the LULCC-induced changes in day-to-day SAT variability and the associated economic impacts:

Day-to-day SAT variability reflects the magnitude of daily temperature fluctuations. Greater SAT variability implies stronger daily heat or/and cold extremes⁴⁸. As shown in Fig. 3d–f, the BGC impact plays a leading role in LULCC-induced SAT variability changes. The SAT variability is decreased at high latitudes in the Northern Hemisphere (north of 60°N) but increased in the tropics and subtropics (30°S–30°N) (Fig. 3f). It has been documented that the warming due to increased atmospheric CO₂ concentrations leads to Arctic sea-ice loss and thus reduces the meridional temperature gradient in the extratropics due to Arctic amplification. This further weakens synoptic wave activities and results in decreased day-to-day SAT variability there^{49,50}. The increased SAT variability in the tropics is mainly caused by soil drying in response to the warming from the increased CO₂ concentrations^{49–51}.

Economic impacts of day-to-day SAT variability changes

The annual mean SAT changes capture a fraction of SAT impacts on economies. In addition to the annual mean SAT changes, the changes in day-to-day SAT variability have a greater economic influence²⁵. With the global maps of annual mean day-to-day SAT variability changes induced by BGC and BGP impacts of historical LULCC and their combination (Fig. 3), we investigate their corresponding influences on economies. Overall, the signs of the BGC and BGP impacts on economies are opposite in many countries (Fig. 8a and c). The combined impact on economies is regulated by the BGC impact due to its dominant role in impacting the day-to-day SAT changes (Figs. 3f and 8e). In comparison with countries in midlatitudes and high-latitudes, low-latitude countries with smaller seasonal temperature variations are damaged more given increases in day-to-day SAT variability because they are more difficult to adapt to greater temperature fluctuations²⁵ (Fig. 8e). In addition, low-latitude countries with poorer economies (Supplementary Fig. 4) are more vulnerable to the negative economic impact of the increased day-to-day SAT variability because of less insurance and risk-management practices²⁵. Therefore, many low-latitude and low-income countries (e.g., the majority of African and Southeast Asian countries) are greatly damaged from the increased day-to-day SAT variability imposed by the combined impact of LULCC (Fig. 8e). In contrast, the economies of many rich countries in the extratropics (e.g., Canada, the US, and Western European countries) benefit from reduced day-to-day SAT variability there. Quantitatively, the 80:20 and 90:10 ratios of the global economic gap are increased by +9.36% and +2.49%, respectively, due to the combined impact of LULCC. Therefore, in addition to the annual mean SAT impact, LULCC-induced day-to-day SAT variability changes further deteriorate global economic inequality.

We further decompose the impact of annual mean day-to-day SAT variability changes on the annual GDP per capita into economic contributions from day-to-day SAT variability changes in four seasons²⁵ (see Methods). It is shown that spring dominates the annual economic impacts in many countries (Supplementary Figs. 5 and 6). This is mainly due to greater economic sensitivities to spring day-to-day SAT variability changes compared with the other three seasons²⁵. In addition, LULCC impacts on spring day-to-day SAT variability are also prominent (Supplementary Fig. 5a). Overall, the total impact of seasonal day-to-day SAT variability changes on annual GDP per capita is comparable to that calculated by annual mean day-to-day SAT variability changes with slightly noticeable differences in the USA, Russia, and Finland (Supplementary Fig. 6e and f).

Lines 510–542 in the section of the Methods (i.e., estimates of economic impacts of day-to-day SAT variability changes):

Estimates of economic impacts of day-to-day SAT variability changes. In addition to annual mean SAT, we also calculate day-to-day SAT variability, which reflects the magnitude of daily temperature fluctuations; greater temperature variability implies stronger heat or/and cold extremes⁴⁸. The annual mean day-to-day SAT variability is calculated as the intra-monthly standard deviation of daily SAT averaged across 12 months of a year²⁵. The measure of day-to-day variability is seasonally adjusted because economic agents are already adapted to the seasonal cycle²⁵. Both heat and cold extremes are involved in day-to-day SAT variability. A recent study has documented a damage function describing country-level responses of the GDP per capita growth rate to every extra degree Celsius of the day-to-day SAT variability²⁵:

$$\Delta Growth_{LULCC} = \alpha \times \Delta TVAR_{LULCC} \quad (8)$$

where $\Delta TVAR_{LULCC}$ is the LULCC-induced country-level day-to-day SAT variability change and $\Delta Growth_{LULCC}$ is the associated change in the growth rate of GDP per capita. α is a negative parameter denoting the decrease in the growth rate of GDP per capita from an extra degree Celsius of day-to-day SAT variability (greater $|\alpha|$ implies greater damage). The parameter is dependent on the country-level seasonal temperature difference. The magnitude of α is greater in countries with smaller seasonal variations (e.g., tropical countries). Using the damage response function and then following the same process as the estimates of economic impacts of annual mean SAT changes, the economic impacts of day-to-day SAT variability changes can be derived with country-level day-to-day SAT variability changes induced by LULCC ($\Delta TVAR_{LULCC}$) calculated from CMIP6 simulations.

In addition to the annual mean seasonally-adjusted day-to-day SAT variability (averaged across 12 months), we also calculate seasonal day-to-day SAT variability averaged across three months of the season for boreal spring (March, April, May), summer (June, July, August), autumn (September, October, November), and winter (December, January, February). Then, the impacts of these seasonal day-to-day SAT variability changes induced by LULCC on annual GDP per capita are investigated according to a season-specific damage function developed by ref²⁵. These seasonal influences show their respective economic contributions to annual GDP per capita changes and can be summed to reflect annual cumulative economic impacts with the seasonal cycle. For a given day-to-day SAT variability, its economic impact varies in different seasons²⁵. Compared with other seasons, spring shows greater economic damage in response to increased day-to-day SAT variability²⁵. Economic growth in winter is insensitive to day-to-day SAT variability changes because economic agents may be sheltered from extreme weather effects by reducing outdoor activities²⁵.

I outline some more specific comments below, but my main concerns are that the authors need to at least estimate the biogeochemical effect of LULCC on GDP in order to make a claim of how LULCC has actually impacted GDP, and that the annual mean temperature effect of LULCC is likely weaker than the seasonal effect, and the impact on economics would likely come from having e.g., a cooler growing season, or a more extreme heat wave, etc.

Reply: Thanks for the valuable comments. In the revised manuscript, we have analyzed the BGC and combined (BGP+BGC) effects of LULCC on economies via altering the annual mean temperature. We have also estimated the changes in the temperature extremes (represented by day-to-day SAT variability) induced by the BGC and BGP impacts of LULCC and the associated economic influences. The results show that the combined impacts of historical LULCC increase global economic inequality via both changes in annual mean SAT and day-to-day SAT variability (Fig. R3e&R5e). Decomposing the economic impacts of day-to-day SAT variability changes into contributions from four seasons, it is shown that spring dominates the annual economic impacts in most countries (Fig. R6). This is mainly due to the greater economic sensitivities to spring day-to-day SAT variability changes compared with the other three seasons (Kotz et al., 2021). In addition, LULCC impacts on spring day-to-day SAT variability are also prominent (Fig. R7a).

References:

Kotz, M., Wenz, L., Stechemesser, A., Kalkuhl, M. & Levermann, A. Day-to-day temperature variability reduces economic growth. *Nat Clim Change* 1–7 (2021)

Specific comments:

- The authors frequently mention the temperature/GDP relationship, but never show the response curve. Since this curve is central to the paper, it should be shown in the paper.

Reply: Thanks for the suggestion. Three representative curves of the bootstrapped temperature–growth response function have been presented in Fig. 4a, respectively showing 25th, 50th, and 75th percentile of 1,000 members of the temperature optimum (Burke et al., 2018). The response function shows that economic growth peaks at the optimal annual mean temperature and declines at higher or lower temperatures (Fig. R8).

References:

Burke, M., Davis, W. M. & Diffenbaugh, N. S. Large potential reduction in economic damages under UN mitigation targets. *Nature* **557**, 549–553 (2018).

Fig. R8. Bootstrapped temperature–growth response function represented by 25th (left dashed curve), 50th (solid curve), and 75th (right dashed curve) percentiles of 1,000 members of the temperature optimum. Vertical lines overlaid on the curves are annual mean temperatures from factual observations with LULCC (brown) and the counterfactual world without the combined BGP and BGC impacts of LULCC (green) for some representative countries.

The corresponding edits have been made in Lines 175–186 in the revision:

Fig. 4 shows the combined impacts of LULCC on economies. For warm countries with annual mean SAT warmer than the temperature optimum (mostly between 30°S and 30°N), LULCC-induced warming makes country-level temperatures deviate further from the temperature optimum, thus damaging the economies there (e.g., Zambia, India, and Saudi Arabia). In contrast, the warming benefits the economic growth for cool countries whose annual mean SAT is colder than the temperature optimum (mostly north of 45°N) because it makes country-level temperatures closer to the temperature optimum (e.g., Iceland and the UK). For mid-latitude countries with annual mean SAT close to the temperature optimum (e.g., the USA and China), economic growth is insensitive to annual mean SAT changes there. These changes are regulated by the BGC

impact of LULCC due to its dominant role in annual mean temperature changes (Supplementary Fig. 2a), except Canada, which is controlled by the BGP-induced cooling (Supplementary Fig. 3a).

- The title, and lines 26-28 of the abstract, are too strong a claim for what the results show – specifically the paper is only capturing the annual mean biogeophysical effects of LULCC, which are only a fraction of the actual LULCC impact on climate

Reply: Thanks for the comment. In the revision, as suggested, we present the BGP and BGC impacts of LULCC on the economy. In addition to annual-mean influences, we discuss the impact of day-to-day SAT variability on the economy. The results show that global economic inequality is increased due to the combined BGP and BGC impacts on annual mean SAT and day-to-day SAT variability. Given that the BGP and BGC impacts have contrasting effects on global economic inequality, we have changed the original title to “Contrasting Influences of Biogeophysical and Biogeochemical Impacts of Historical Land Use on Global Economic Inequality”. With updated results, the abstract has been substantially revised.

- Figure 1: I know total area of urban land is small compared to the other land cover types in this figure, but the fact that it shows up on the legend but isn't visible at all in the plot is a little weird. I'm not sure what the authors should do about this, but one option would be to put write the total area at both the start and end of the time series in the legend next to each colour, or right on the plot, so the reader can see "ah, this tiny number for urban area, but the increase over the time considered is really big!" (Even though they can't see the line on the figure).

Reply: Thanks for the comment. We have made adjustments to Fig. 1a to see the fraction of urban land more clearly (Fig. R9).

Fig. R9. Global historical LULCC from 1850 to 2014. Land-cover fractions (relative to global land except for Antarctica) of primary vegetation (dark green), secondary vegetation (light green), cropland (orange), grazing land (yellow), and urban land (pink). The fractions of the five land-cover types add up to less than 1 due to the coverage of ice and water (shown in the white area).

- Line 112: “America” – either change to “United States of America” if you mean the country, or (from the figure, it looks like much more than just the USA), “North America” / “from southern Canada down through Central America” / “The Americas” if you also want to include the changes in South America as part of this sentence, though the authors specifically call out the Amazon.

Reply: “America” has been changed to “the USA” in Lines 130–132 in the revision:

The expansion of grazing land was more significant, mainly located in the USA, south of the Amazon, central and southern Africa, central Asia, and Australia.

- Line 145: “America” -> “USA” or “United States of America”, “French” -> “France”

Reply: Done.

- Line 147: I find this usage of brackets, here and several other places in the paper, really difficult to read – and I’m not alone! See for example:
<https://eos.org/opinions/parentheses-are-not-for-references-and-clarification-saving-space>
This is a stylistic comment though, and the authors can choose to ignore it. But it does make the paper harder to read.

Reply: Thank you very much for the comment. In the revision, we have removed most parentheses, just retaining a few for clarification.

- Line 150: I don’t think this is verified with observational constraints, or I’m missing something. It is somewhat impossible to verify the main idea in this study with observational constraints – rather, what is shown in this paragraph is just applying the temperature/GDP relationship to observed, rather than modelled, temperatures.

Reply: We compare country-level SAT and economic conditions between the two worlds: one is the factual world with LULCC, and the other is the counterfactual world, which is identical to the former except without LULCC. Given the existing SAT biases in CMIP6, the SAT differences derived from the “historical” and “hist-noLu” experiments are used only. We use reanalysis data to represent the SAT in the factual world. Applying the simulated LULCC-induced SAT differences to the reanalysis, we obtain the SAT in the counterfactual world without LULCC. This approach of bias correction is called the “delta” method, which has been widely used in previous studies (Differbaugh and Burke, 2019). Similarly, socioeconomic statistical data are used to represent the economic condition of the factual world. Accordingly, the counterfactual economies without LULCC are estimated using the factual statistics minus the LULCC-induced economic differences with the “delta” method. The estimation of the counterfactual world is with observational constraints because its SAT and economic conditions are based on factual observations rather than modeled. Due to the observational constraints, the trends of SAT and economic growth in the counterfactual world are similar to those in the observations during the study period (Fig. 4c). In the revision, to avoid confusion, we have made corresponding edits in Lines 94–101 in the introduction:

In summary, the country-level SAT and economic conditions of the two worlds are compared: one is the factual world with LULCC, and the other is the counterfactual world, which is identical to the former except without the BGP and BGC impacts of LULCC. The factual world is represented by SAT reanalysis and economic statistics^{38–40}. Grid-cell SAT is aggregated to the country level, weighted by population distribution in the country⁴¹. The counterfactual world without LULCC is estimated with observational constraints, calculated as the observations minus modeled LULCC-induced differences.

Lines 422–436 in the Methods of “Factual and counterfactual worlds”:

Factual and counterfactual worlds. We compare country-level SAT and economic conditions between two worlds: one is the factual world with LULCC, and the other is the counterfactual world, which is identical to the former except without the BGP and BGC impacts of LULCC. Given the existing SAT biases in CMIP6, the SAT differences induced by LULCC (ΔT_{LULCC}) derived from the “historical” and “hist-noLu” experiments are used only. We use reanalysis data to represent the SAT in the factual world (T_{Obs}). Applying the simulated LULCC-induced SAT difference to the reanalysis, we obtain the SAT in the counterfactual world without LULCC ($T_{NoLULCC}$):

$$T_{NoLULCC} = T_{Obs} - \Delta T_{LULCC} \quad (1)$$

This approach of bias correction is called the “delta” method, which has been widely used in previous studies²⁸. Similarly, socioeconomic statistical data are used to represent the economic conditions of the factual world. Accordingly, the counterfactual economies without LULCC are estimated using the factual statistics minus the LULCC-induced economic differences with the “delta” method (see estimates of economic impacts of annual mean SAT changes below). Thus, the calculation of SAT and economies in the counterfactual world is constrained by observations.

References:

Diffenbaugh, N. S., Burke, M. Global warming has increased global economic inequality. *Proc Natl Acad Sci U S A* **116**, 9808-9813 (2019).

- *Figure 5: It would be helpful to give subplots titles. Also, the blue/red shading of bars is rather confusing. Is the colour split at the global median per capita GDP? if so, state. If not, and the split is arbitrary, consider using a continuous (rather than diverging) colour bar*

Reply: Thanks for the comment. Fig. 5c in the original manuscript has been moved to Fig. 6 in the revision. According to the World Bank economic data, the global mean GDP per capita in 2012 was approximately 10000 US\$₂₀₁₀, which is used to distinguish wealthier and poorer countries and split blue/red shading of bars in Fig. 6. Countries with GDP per capita greater than 10000 US\$₂₀₁₀ are grouped into wealthier countries (in red). Otherwise, they are grouped into poorer countries (in blue). Subplot titles have been added in Fig. 5. The blue/red shadings of bars in Fig. 6 have been clarified in Lines 805–808:

Blue/red color is split at the global GDP per capita in 2012 (approximately 10000 US\$₂₀₁₀)³⁹. Countries with GDP per capita greater than 10000 US\$₂₀₁₀ are grouped into wealthier countries (in red). Otherwise, they are grouped into poorer countries (in blue).

- Line 186-188: it looks, from this figure, like a lot of wealthy countries also benefited – the area under the combined USA/Germany red bar is quite large (hard to tell from this figure if it is larger than the negative red area in the Canada/Russia bars).

Reply: The figure mentioned here is current Fig. 6a, illustrating the individual BGP impact on economies in countries of different economic conditions. Note that the width of the bar in the figure for each country just indicates the fractional contribution of the country to global total GDP in 2012. To identify whether a country benefits or damages from LULCC, we need to see the height of the bar on the y-axis (GDP per capita changes induced by LULCC). Countries are sorted by their relative changes in GDP per capita in 2012 induced by LULCC, from damages on the left to benefits on the right. For the BGP impact, the USA and Germany, with large contributions to global GDP in 2012, feature slight increases in GDP per capita, which, however, are much smaller than the decreases in Canada and Russia. Fig. 6a qualitatively illustrates that the BGP cooling impact of LULCC harms the economies of many richer countries mostly in a cool climate (bars shaded in the redder color on the left) but favors many poorer countries in the warm tropics (bars shaded in the bluer color on the right). We further use common measures of economic inequality (reflecting the global economic gap between the top wealthy country and the bottom poor country) to quantitatively estimate the impacts of LULCC on global economic inequality. The results further confirm the reduced global economic equality from the BGP impact (Fig. 7a and d). Corresponding edits have been made in Lines 214–221 in the revision:

The individual BGP and BGC impacts and their combination on economies in countries of different economic conditions are illustrated in Fig. 6. The combined impact of LULCC largely modulated by the BGC warming benefits many richer countries with higher GDP per capita (bars shaded in the redder color on the right), but damages many poorer countries with lower GDP per capita (bars shaded in the bluer color on the left). In contrast, the BGP cooling impact of LULCC, albeit canceled out by the BGC-dominated warming, harms the economies of many richer countries (e.g., Russia, Canada, and Norway) mostly in cool climates but favors many poorer countries (e.g., Indonesia, Egypt, and India) in the warm tropics.

- Figure 6: needs a legend, and a more descriptive caption - what are the two different box/whiskers in panel b? I assume one is the change with and one without land use, but add a legend to panel b also. (It is confusing, as the colours match (a), but the line styles don't, so it isn't clear if (a)'s legend also applies to (b)).

Reply: Thanks for the suggestion. Fig. 6 is current Fig. 7. Fig. 7 now shows the impact of LULCC on global economic inequality using two common measures of economic inequality: 80:20 and 90:10 ratios of the population-weighted percentiles of GDP per capita, represented by dark and light blue, respectively. Both the 80:20 and 90:10 ratios reflect the global economic gap between the top wealthy country and the bottom poor country. Panels a–c in the figure show time series of the 25th–75th percentile range of 80:20 and 90:10 ratios in the counterfactual world without LULCC, both with corresponding observations in the factual world with LULCC (black dotted line). Panels d–f show relative changes in both 80:20 and 90:10 ratios induced by LULCC, matching colors in panel a. The black box outside panel f on the right is to illustrate the distribution of percentiles of box-and-whiskers. The legend and caption in Fig. 7 in

the revision have been updated following the suggestion.

- *As mentioned above, the temperature/GDP response function needs to be shown somewhere in the paper.*

Reply: Thanks for your suggestion. Three representative curves of the bootstrapped temperature–growth response function have been presented in Fig. 4a, respectively showing 25th, 50th, and 75th percentiles of 1,000 members of the temperature optimum. The response function shows that economic growth peaks at the optimal annual mean temperature and declines at higher or lower temperatures.

- *Line 229: “It is because that a within” – grammar?*

Reply: Corrected.

- *Line 232: I’m glad the authors discuss the limitations of averaging climate (and GDP) over large countries.*

Reply: Thanks for the positive comment.

- *Line 234: the authors say here that they weight the climate response by population density within each country, but I can’t find a description of how this was done in the methods.*

Reply: Thanks for the suggestion. In the revision, we have added a section “Spatial country-level aggregation” to the Methods. Grid-cell temperature fields are aggregated to the country level, weighted by population distribution in the country according to the gridded population data from the Gridded Population of the World version 4 (GPWv4). The aggregated country-level temperature reflects the representative temperature where the majority of populations and economic production of the country are situated; temperature changes in sparsely populated areas with less economic activities contribute to less economic impacts in the country. The edits have been made in Lines 415–420:

Spatial country-level aggregation. Grid-cell SAT is aggregated to the country level, weighted by population distribution in the country according to the gridded population data from the Gridded Population of the World version 4 (GPWv4)⁴¹. The aggregated country-level SAT reflects the representative SAT where the majority of populations and economic production of the country are situated; temperature changes in sparsely populated areas with less economic activities contribute to less economic impacts in the country.

- *Line 360: I assume the bootstrapped temperature-growth functions are not capturing different sensitivities to seasonality and extremes, so what extra information are they giving you?*

Reply: The extra information we want to convey is that we have accounted for the uncertainty of the temperature–growth response function. A total of 1,000 members of the response function (generated by bootstrapping) are applied. Thus, a country-level population-weighted SAT change induced by LULCC corresponds to 1,000 responses

of GDP per capita growth. The set of response functions shows 1,000 cases of the “temperature optimum” of economic growth. For the seasonality and extremes, please see the response to the major concerns.

- The authors do, in the last few paragraphs of the paper, mention both the carbon cycle effects of LULCC and the seasonal impact of LULCC, but they don't attempt to estimate or quantify these factors, which seems critical to actually make any claims about the historical LULCC impact on economic inequality (see main concerns at the start of my comments).

Reply: Please see the response to the major concerns.

Reply to the comments by Reviewer #3

We thank the reviewer for the comments and suggestions to improve our manuscript. Below are our point-by-point responses to these comments. The reviewer's comments are in italics, our responses are in normal font, and manuscript revisions are in blue.

This manuscript presents results from a study that considers climate impacts from historical land-use change over the period 1991-2010 and the resulting implications for the economic well-being of different countries in the world. They specifically consider the biogeophysical impacts of land-use change on annual mean surface air temperatures and then use a temperature-growth response function to estimate whether per capita GDP in each country is likely to increase or decrease as a result of that change in temperature. By comparing all countries across the world the authors determine that historical land-use is likely to have led to a reduction in the global income gap between the 80th and 20th percentiles of the global population. This is a very interesting, novel, and thought-provoking study. The methodology is sound and appears to be reproducible. Supporting evidence is provided for all conclusions and it is well-written. It is worthy of publication, subject to consideration of a few comments and suggestions, as detailed below:

Reply: Thanks for noting the novelty of this study and the positive comments.

• LUH2 is a land-use product not a land cover product. The authors should be careful not to refer to “land cover” when describing the LUH2 historical land-use products in the Methods section.

Reply: Thanks for the suggestion. It has been corrected in Lines 337–338:

Historical land-use data. Land-use forcing for the “historical” experiment (from 1850 to 2014) in CMIP6 is based on the Land-Use Harmonization 2 (LUH2) project^{4,43,46}.

• Line 276: secondary vegetation is “re-growing” vegetation, not “regrown”.

Reply: Corrected in Lines 340–342:

Primary and secondary vegetation are both natural vegetation (either forest or non-forest), but primary vegetation has not been disturbed by humans, while secondary vegetation is re-growing vegetation recovering from previous human disturbance.

• The term “America” as a country name should be replaced with “USA” throughout the manuscript.

Reply: Done.

• In the section beginning on line 102 entitled “Historical LULCC since the Industrial Revolution”, and in Figures 1 & 2, the presented results appear to be from Hurtt et al. 2020 (reference number 4 in the reference list). However, although Hurtt et al. 2020 is cited early in the Introduction, it is not cited at all in this section or in the captions for Figures 1 & 2. In addition to citing Hurtt et al. 2020 in this section, the LUH2 dataset

itself should also be cited since it is being used extensively here and has its own DOI.

Reply: Thanks for pointing this out. Figs. 1–2 refer to Figures in Hurtt et al. (2020), plotted using the LUH2 dataset. In the revision, Hurtt et al. (2020) has been cited in Lines 122–124 in the section “Historical LULCC since the Industrial Revolution”, where the LUH2 dataset (Hurtt et al., 2019) has been cited as well:

Fig. 1 shows the global historical LULCC from 1850 to 2014 based on the Land-Use Harmonization 2 (LUH2) dataset, which is the land-use forcing for the CMIP6 “historical” experiment^{4,43,46}.

References:

Hurtt, G. C. et al. Harmonization of global land use change and management for the period 850–2100 (LUH2) for CMIP6. *Geosci Model Dev* **13**, 5425–5464 (2020).
Hurtt, G. C. et al. Harmonization of Global Land Use Change and Management for the Period 850-2015. Version 20211004[1]. *Earth System Grid Federation*. (2019) <https://doi.org/10.22033/ESGF/input4MIPs.10454>.

• The analysis is based on several simplifying assumptions that allow the study to consider (in isolation from other factors) a specific response to a specific climate forcing. This is a very useful approach, and (as the authors point out) it facilitates understanding of the mechanisms of economic change in response to land-use change, and could potentially guide future land-use policies. On the other hand, it is also useful to place this study into the broader context by also discussing some of the factors not directly considered here. For example, what are impacts of land-use change on other climatic variables such as precipitation and what is the response of economies to those changes? Also, should we also consider the more direct economic impacts of land-use change (such as loss of ecosystem services, or the gain of agricultural commodities)? The authors do mention these simplifying assumptions up front, but some additional discussion in the “Implications and Uncertainties” section towards the end of the manuscript could be helpful.

Reply: Thanks for the valuable suggestion. Historical LULCC has significant impacts on precipitation as well. Nonetheless, no significant effects of precipitation on country-level GDP per capita were found (Burke et al., 2018; Pretis et al., 2018; Kotz et al., 2021). Some analyses show that the precipitation impacts on within-country regional economic production are noticeable (Fishman, 2016). We have discussed these in Lines 316–323 in the revision:

LULCC has significant impacts on precipitation as well^{12,33}. Nonetheless, no significant effects of precipitation on country-level GDP per capita were found^{25–27}, although some analyses show that precipitation impacts on within-country regional economic production are noticeable^{58–60}. The country-level ability to resist risks of climate change is greater than the within-country regional level because adaptation is likely to be coordinated among different regions at the country level²⁵. This implies the necessity of within-country regional analyses in countries covering large areas, particularly for analyzing the economic impacts imposed by climate factors that are strongly spatially heterogeneous.

It has been documented that long-term economic sustainable development is limited

due to substantial disruption to the ecosystem from historical LULCC despite short-term regional economic growth with the gains of agricultural/industrial commodities (Goldstein et al., 2012; Ouyang et al., 2020). This study reveals the potential declines in global economic equitable and sustainable development due to climate feedback of LULCC via biogeophysical and biogeochemical processes. The extent to which society should limit its impact on the ecosystem to prevent long-term economic damages with short-term benefits poses a difficult challenge for policymakers. The solution hinges on estimating overlooked potential long-term economic damages behind short-term economic growth. A sound understanding of socio-climatic interactions is vital for assessing where the optimal balance of global economic development and environmental protection lies. In the revision, related discussions have been provided in Lines 80–89 in the introduction:

It has been documented that long-term economic sustainable development is limited due to substantial disruption to the ecosystem from historical LULCC despite short-term regional economic growth with the gains of agricultural/industrial commodities^{34–37}. Nonetheless, the economic impact of historical LULCC through climate feedback is still unclear. This is because the climatic impacts of LULCC on economies are much more complicated than those of greenhouse gases and anthropogenic aerosols. LULCC has both BGP and BGC impacts on climate, and the BGP impacts vary in latitudes and land surface types. Furthermore, besides the economic effects of the annual-mean temperature change induced by historical LULCC, the additionally imposed effects on economies from day-to-day temperature variability changes need to be figured out as well.

Lines 324–331 in the “Discussion”:

Despite short-term regional economic growth brought by land-use activities with gains of agricultural/industry commodities, this study reveals the potential declines in global economic equitable and sustainable development due to climate feedback of LULCC. The extent to which society should limit its impact on the ecosystem to prevent long-term economic damages with short-term benefits poses a difficult challenge for policymakers. The solution hinges on estimating overlooked potential long-term economic damages behind short-term economic growth. A sound understanding of socio-climatic interactions is vital for assessing where the optimal balance of global economic development and environmental protection lies.

References:

- Burke, M., Davis, W. M. & Diffenbaugh, N. S. Large potential reduction in economic damages under UN mitigation targets. *Nature* **557**, 549–553 (2018).
- Kotz, M., Wenz, L., Stechemesser, A., Kalkuhl, M. & Levermann, A. Day-to-day temperature variability reduces economic growth. *Nat Clim Change* 1–7 (2021) doi:10.1038/s41558-020-00985-5.
- Pretis, F., Schwarz, M., Tang, K., Haustein, K. & Allen, M. R. Uncertain impacts on economic growth when stabilizing global temperatures at 1.5C or 2C warming. *Philosophical Transactions Royal Soc Math Phys Eng Sci* **376**, 20160460 (2018).
- Fishman, R. More uneven distributions overturn benefits of higher precipitation for crop yields. *Environ Res Lett* **11**, 024004 (2016).
- Ouyang, Z. et al. Using gross ecosystem product (GEP) to value nature in decision making. *Proc National Acad Sci* **117**, 14593–14601 (2020).

Goldstein, J. H. et al. Integrating ecosystem-service tradeoffs into land-use decisions. *Proc National Acad Sci* **109**, 7565–7570 (2012).

• *Change in per capita gross domestic product is the measure of economic impact chosen for use in this study. However, it provides an average estimate across all citizens within a country, and in many nations there is often a wide disparity between the wealthiest citizens and the poorest. As the authors do point out, subnational estimates of per capita GDP are difficult to obtain, but I am curious whether there are estimates of the spread of income within countries that could be used to estimate within-country economic inequalities and provide better estimates of the resulting global economic inequalities. Or are there measures of economic activity other than GDP that could be used here? Perhaps this is something that could be touched upon in the Discussion section as well.*

Reply: Thanks for the helpful comments. With all the influencing factors (e.g., trade globalization), global economic inequality over the past decades has been mitigated (Alvaredo et al., 2017; Hammar et al., 2020). Decomposing the declining economic inequality into contributions from between- and within-country, it has been found that the economic gap between developed and developing countries accounts for four-fifths of the declining trend of global inequality (Hammar et al., 2020). As the trend of global inequality is dominated by the between-country economic gap, the analysis is performed at the country level. The results show that the climate feedback of historical LULCC negatively impacts the decreasing trend of country-level global economic inequality. Country-level coordinated adaptation leads to greater resilience to climate change. Attention should also be paid to the within-country regional economic impact under climate change because of higher regional climate vulnerability (Kotz et al., 2021). However, a global within-country analysis is challenging owing to insufficient global spatiotemporal subnational socioeconomic data. To overcome this issue as much as possible, we aggregate temperature to the country level by population-weighting in the country. This provides some indications of the within-country regional economic gap; temperature changes in sparsely populated areas with less economic activities contribute to less economic impacts in the country. Some economic inequality analyses were based on household income or consumption data according to national statistical surveys (Alvaredo et al., 2017). Recent studies show that poor people are losing more relative to wealthy people when exposed to extreme heat (Hallegatte and Rozenberg, 2017). Therefore, global warming also deteriorates economic inequality at the individual or household level (Hallegatte and Rozenberg, 2017), in addition to the country level. We have discussed these in Lines 297–315 in the revision:

With all the influencing factors (e.g., trade globalization), global economic inequality over the past decades has been mitigated^{55,56}. Decomposing the declining economic inequality into contributions from between- and within-country, it has been found that the economic gap between developed and developing countries accounts for four-fifths of the declining trend of global inequality⁵⁵. As the trend of global inequality is dominated by the between-country economic gap, the analysis is performed at the country level. The climate impacts of historical LULCC negatively impact the decreasing trend of country-level global economic inequality. Country-level coordinated adaptation leads to greater resilience to climate change. Attention should also be paid to the within-country regional economic impact under climate change because of higher regional climate vulnerability²⁵. However, a global within-country

analysis is challenging owing to insufficient global spatiotemporal subnational socioeconomic data^{25,39}. To overcome this issue as much as possible, we aggregate temperature to the country level by population weighting in the country. This provides some indications of the within-country regional economic gap; temperature changes in sparsely populated areas with less economic activities contribute to less economic impacts in the country. Some previous analyses on economic inequality were based on household income or consumption data according to national statistical surveys^{39,55,56}, showing that poor people are losing more relative to wealthy people when exposed to extreme heat⁵⁷. Therefore, global warming induced by the combined effects of historical LULCC also deteriorates economic inequality at the individual or household level⁵⁷, in addition to the country level.

References:

- Alvaredo, F., Chancel, L., Piketty, T., Saez, E. & Zucman, G. ‘World inequality report 2018’, WID.world (2017).
- Hammar, O. & Waldenström, D. Global Earnings Inequality, 1970–2018. *Econ J* **130**, 2526–2545 (2020).
- Kotz, M., Wenz, L., Stechemesser, A., Kalkuhl, M. & Levermann, A. Day-to-day temperature variability reduces economic growth. *Nat Clim Change* 1–7 (2021) doi:10.1038/s41558-020-00985-5.
- Hallegatte, S. & Rozenberg, J. Climate change through a poverty lens. *Nat Clim Change* **7**, 250–256 (2017).

REVIEWER COMMENTS, second round

Reviewer #1 (Remarks to the Author):

The authors have made a lot of improvements according to the previous comments and all my concerns have been addressed.

Reviewer #2 (Remarks to the Author):

I commend the authors on their revision of the manuscript, in particular for incorporating the biogeochemical effects of LULCC into their analysis, and for evaluating the influence of changes in temperature variability (rather than only annual mean temperature).

I have only a handful of very minor comments, below:

- Line 295: 1993-2012 isn't the past two decades. Maybe say "Over the 2 decades considered here" instead.

- Figure 2 / 4a has very small font and is difficult to read

- Line 361 / CMIP6 Methods section: it would be good to add the caveat that using the 1pctCO2 simulation warming at 9% increased CO2 doesn't *exactly* isolate the CO2 effect because the 1pctCO2 simulations aren't in equilibrium. I think that is fine for this study - there is not a perfect solution within the CMIP6 family of simulations, but the authors should just mention that caveat for completeness.

Reviewer #3 (Remarks to the Author):

Review of "Contrasting Influences of Biogeophysical and Biogeochemical Impacts of Historical Land Use on Global Economic Inequality" by S. Liu et al.

This manuscript examines the national-level biogeophysical and biogeochemical impacts of land-use change on climate/temperature between 1993 to 2012 (resulting from land-use change since 1850). It then goes on to consider the national economic impact resulting from these temperature changes. The overall conclusion is that although the biogeophysical impacts of land-use change could help to reduce global economic inequality, the larger biogeochemical impacts of land-use change are likely to increase global economic inequality.

This manuscript is a resubmission after a previous review. The authors have done a good job of addressing the previous reviewer comments and suggestions. As a result, the new manuscript is greatly improved, and includes a lot of additional new analysis. The central concept of the study is still interesting and novel, although the main conclusion of the study is now changed due to the suggested consideration of biogeochemical impacts as well as biogeophysical impacts of land-use change.

I have just a couple of suggestions for improvements:

1) Several of the figures appear to present data taken directly from other datasets, products, or papers and adapted for use here (for example Figure 1 and Figure 2 both present data from the LUH2 dataset). The original sources are cited in the text but my preference would be to see them cited in the figure captions as well.

2) The underlying concept of the economic response to temperature (the temperature-growth response function) seems to be based on a somewhat steady-state scenario (i.e. relatively small changes in temperature and economic productivity around a stable mean). What doesn't appear to

be factored in or discussed are the economic disruptions and transitions that can be caused by a changing climate. For example, if specific crops are no longer viable in a new climate, there will be a (possibly transient) economic disruption while agricultural lands are converted to new crop types. These more disruptive economic events that result from temperature change could be discussed in the Discussion with some statements about how this could possibly modify the results (if at all).

Reply to the comments by Reviewer #1

We thank the reviewer for the comments and suggestions to improve our manuscript. Below are our point-by-point responses to these comments. The reviewer's comments are in italics and our responses are in normal font.

The authors have made a lot of improvements according to the previous comments and all my concerns have been addressed.

Reply: Thanks for the reviewer's valuable suggestions/comments which substantially improved the manuscript.

Reply to the comments by Reviewer #2

We thank the reviewer for the comments and suggestions to improve our manuscript. Below are our point-by-point responses to these comments. The reviewer's comments are in italics, our responses are in normal font, and manuscript revisions are in blue.

I commend the authors on their revision of the manuscript, in particular for incorporating the biogeochemical effects of LULCC into their analysis, and for evaluating the influence of changes in temperature variability (rather than only annual mean temperature).

Reply: Thanks for the reviewer's valuable suggestions/comments which substantially improved the manuscript.

I have only a handful of very minor comments, below:

- Line 295: 1993-2012 isn't the past two decades. Maybe say "Over the 2 decades considered here" instead.

Reply: Corrected in Lines 286–288 in the revision:

This implies that the impact of LULCC on the global economy since the Industrial Revolution is greater than the impact accumulated just over the two decades considered here.

- Figure 2 / 4a has very small font and is difficult to read

Reply: The font has been adjusted in these Figures.

*- Line 361 / CMIP6 Methods section: it would be good to add the caveat that using the 1pctCO2 simulation warming at 9% increased CO2 doesn't *exactly* isolate the CO2 effect because the 1pctCO2 simulations aren't in equilibrium. I think that is fine for this study - there is not a perfect solution within the CMIP6 family of simulations, but the authors should just mention that caveat for completeness.*

Reply: Thanks for the comments. The caveat has been added in Lines 397–400 in the revised manuscript:

Note that the 1pctCO2 simulation is not in equilibrium for the increasing CO2 concentrations. It is a compromise to estimate the CO2 effect here because there is currently not a set of CMIP6 experiments that allows us to directly evaluate the BGC impact of historical LULCC.

Reply to the comments by Reviewer #3

We thank the reviewer for the comments and suggestions to improve our manuscript. Below are our point-by-point responses to these comments. The reviewer's comments are in italics, our responses are in normal font, and manuscript revisions are in blue.

Review of “Contrasting Influences of Biogeophysical and Biogeochemical Impacts of Historical Land Use on Global Economic Inequality” by S. Liu et al.

This manuscript examines the national-level biogeophysical and biogeochemical impacts of land-use change on climate/temperature between 1993 to 2012 (resulting from land-use change since 1850). It then goes on to consider the national economic impact resulting from these temperature changes. The overall conclusion is that although the biogeophysical impacts of land-use change could help to reduce global economic inequality, the larger biogeochemical impacts of land-use change are likely to increase global economic inequality.

This manuscript is a resubmission after a previous review. The authors have done a good job of addressing the previous reviewer comments and suggestions. As a result, the new manuscript is greatly improved, and includes a lot of additional new analysis. The central concept of the study is still interesting and novel, although the main conclusion of the study is now changed due to the suggested consideration of biogeochemical impacts as well as biogeophysical impacts of land-use change.

Reply: Thanks for the reviewer's valuable suggestions/comments which substantially improved the manuscript.

I have just a couple of suggestions for improvements:

1) Several of the figures appear to present data taken directly from other datasets, products, or papers and adapted for use here (for example Figure 1 and Figure 2 both present data from the LUH2 dataset). The original sources are cited in the text but my preference would be to see them cited in the figure captions as well.

Reply: Thanks for the suggestion. We have cited them in the captions of Figs. 1&2 as well.

2) The underlying concept of the economic response to temperature (the temperature-growth response function) seems to be based on a somewhat steady-state scenario (i.e. relatively small changes in temperature and economic productivity around a stable mean). What doesn't appear to be factored in or discussed are the economic disruptions and transitions that can be caused by a changing climate. For example, if specific crops are no longer viable in a new climate, there will be a (possibly transient) economic disruption while agricultural lands are converted to new crop types. These more disruptive economic events that result from temperature change could be discussed in the Discussion with some statements about how this could possibly modify the results (if at all).

Reply: In the temperature-growth response function, the GDP per capita, including economic activities from agriculture, industry, and other sectors, was assembled over the period from 1960 to 2010 (Burke et al. 2018). Over the five decades with changing

climate, the possible economic disruptions and transitions should be reflected in the assembled economic data and thus are implicitly accounted in the function (i.e., the distinct slope of economic response to the same temperature change in different temperature ranges, as shown in Fig. 4a).

Some edits have been made in Lines 446–448, 460–461, and 323–325:

Recent research assembled historical climatic and economic data (covering 165 countries from 1960 to 2010) and built the empirical relationship of country-level annual mean temperature and GDP per capita growth rate^{27,28}.

As background temperature deviates from the optimum, economic production gradually becomes more sensitive to temperature disturbance.

Historical warming induced by anthropogenic activities has caused economies of poor countries to be more vulnerable to climate change. In the warmer future, economies will be damaged more seriously from warming.

References:

Burke, M., Davis, W. M. & Diffenbaugh, N. S. Large potential reduction in economic damages under UN mitigation targets. *Nature* **557**, 549–553 (2018).